# GRADIENT IMPORTANCE LEARNING FOR INCOMPLETE OBSERVATIONS

**Qitong Gao**[*]    **Dong Wang**[*]    **Joshua D. Amason**[*]    **Siyang Yuan**[*]    **Chenyang Tao**[*]

**Ricardo Henao**[*]    **Majda Hadziahmetovic**[*]    **Lawrence Carin**[*,†]    **Miroslav Pajic**[*]

## ABSTRACT

Though recent works have developed methods that can generate estimates (or imputations) of the missing entries in a dataset to facilitate downstream analysis, most depend on assumptions that may not align with real-world applications and could suffer from poor performance in subsequent tasks such as classification. This is particularly true if the data have large missingness rates or a small sample size. More importantly, the imputation error could be propagated into the prediction step that follows, which may constrain the capabilities of the prediction model. In this work, we introduce the gradient importance learning (GIL) method to train multilayer perceptrons (MLPs) and long short-term memories (LSTMs) to *directly* perform inference from inputs containing missing values *without imputation*. Specifically, we employ reinforcement learning (RL) to adjust the gradients used to train these models via back-propagation. This allows the model to exploit the underlying information behind *missingness patterns*. We test the approach on real-world time-series (*i.e.*, MIMIC-III), tabular data obtained from an eye clinic, and a standard dataset (*i.e.*, MNIST), where our *imputation-free* predictions outperform the traditional *two-step* imputation-based predictions using state-of-the-art imputation methods.

## 1 INTRODUCTION

We consider learning from incomplete datasets. Components of the data could be missing for various reasons, for instance, missing responses from participants, data loss, and restricted access issues. This phenomenon is prevalent in the healthcare domain, mostly in the context of electronic health records (EHRs), which are structured as patient-specific irregular timelines with attributes, *e.g.*, diagnosis, laboratory tests, vitals, thus resulting in high missingness across patients for any arbitrary time point. Such missingness introduces difficulties when developing models and performing inference in real-world applications such as Kam & Kim (2017); Scherpf et al. (2019). Existing works tackle this problem by either proposing imputation algorithms (IAs) to explicitly produce estimates of the missing data, or by imposing imputation objectives during inference, *e.g.*, withholding observed values as being the ground-truth and learning to impute them. However, some of these either require additional modeling assumptions for the underlying distributions (Bashir & Wei, 2018; Fortuin et al., 2020), or formatting of the data (Yu et al., 2016; Schnabel et al., 2016). Other applications depend on domain knowledge for pre-processing and modeling such as Yang et al. (2018); Kam & Kim (2017), or introduce additional information-based losses (Cao et al., 2018; Lipton et al., 2016), which are usually intractable with real-world data. Moreover, generative methods (Mattei & Frellsen, 2019; Yoon et al., 2018) could result in imputations with high variation (*e.g.*, low confidence of the output distribution), when data have high missingness rates or small sample sizes. Figure 1 illustrates imputations generated by two state-of-the-art deep generative models on the MNIST dataset with 50%, 70% and 90% of the pixels set as missing (*i.e.*, masked out). It is observed that both approaches suffer from inaccurate reconstruction of the original digits as the missingness rate increases, which is manifested as some of the imputed images not being recognizable as digits. Importantly, the error introduced by the imputation process can be further propagated into downstream inference and prediction stages.

Imputing the missing data (either explicitly or implicitly) is, in most cases, not necessary, more so, considering that sometimes *where* and *when* the data are missing can be intrinsically informative.

---

[*]Duke University, USA. Contact: {qitong.gao, miroslav.pajic}@duke.edu.
[†]King Abdullah University of Science and Technology, Saudi Arabia.

Code available at `https://github.com/gaoqitong/gradient-importance-learning`.

Consider a scenario in which two patients, A and B, admitted to the intensive care unit (ICU) suffer from bacterial and viral infections, respectively, and assume that the healthcare provider monitors the status of the patients by ordering two (slightly) different blood tests periodically, namely, a culture test/panel specific to bacterial infections and a RT-PCR test/panel specific to viral infections. Hence, patient A is likely to have much fewer orders (and results) for viral tests. In both cases, the *missingness patterns* are indicative of the underlying condition of both patients. Moreover, such patterns are more commonly found in incomplete data caused by missing at random (MAR) or missing not at random (MNAR) mechanisms, which usually introduce additional difficulties for inference (Little & Rubin, 2019). Inspired by this, we propose the gradient importance learning (GIL) method, which facilitates an *imputation-free* learning framework for incomplete data by simultaneously leveraging the observed data and the information underlying missingness patterns.

Specifically, we propose to use an *importance* matrix to weight the gradients that are used to update the model parameters in the back-propagation process, *after* the computational graph of the model is settled, which allows the model to exploit the information underlying missingness without imputation. However, these gradients cannot be tracked by automated differentiation tools such as Tensorflow (Abadi et al., 2016). So motivated, we propose to generate the importance matrix using a reinforcement learning (RL) policy *on-the-fly*. This is done by conditioning on the status of training procedure characterized by the model parameters, inputs and model performance at the current training step.



Figure 1: MNIST digits imputed by state-of-the-art imputation methods MIWAE (Mattei & Frellsen, 2019), GAIN (Yoon et al., 2018).

Concurrently, RL algorithms are used to update the policy by first modeling the back-propagation process as an RL environment, or a Markov decision process (MDP). Then, by interacting with the environment, the RL algorithm can learn the optimal policy so the importance matrix can aid the training of the prediction models to attain desirable performance. Moreover, we also show that our framework can be augmented with feature learning techniques, *e.g.*, contrastive learning as in Chen et al. (2020), which can further improve the performance of the proposed imputation-free models.

The technical contributions of this work can be summarized as follows: ($i$) to the best of our knowledge, this is the first work that trains deep learning (DL) models to perform accurate *imputation-free* predictions with missing inputs. This allows models to effectively handle high missing rates, while significantly reducing the prediction error compared to existing imputation-prediction frameworks. ($ii$) Unlike existing approaches that require additional modeling assumptions or that rely on methods (or pre-existing knowledge) that intrinsically have advantages over specific types of data, our method *does not require any* assumptions or domain expertise. Consequently, it can consistently achieve top performance over a variety of data and under different conditions. ($iii$) The proposed framework also facilitates feature learning from incomplete data, as the importance matrix can guide the hidden layers of the model to capture information underlying missingness patterns during training, which results in more expressive features, as illustrated in Figure 3.

## 2 GRADIENT IMPORTANCE LEARNING (GIL)

In this section, we introduce GIL; a method that facilitates training of imputation-free prediction models with incomplete data. In Section 2.2, we show how the gradient descent directions can be adjusted using the *importance matrix* **A**. In Section 2.3, we introduce an RL framework to generate the elements of **A** which can leverage the information underlying missingness during training, as illustrated in Figure 2.

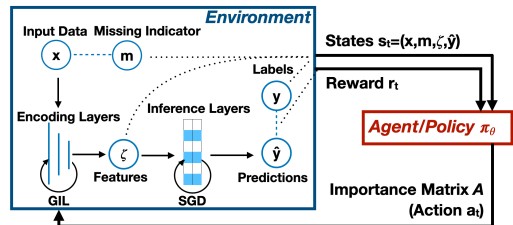

Figure 2: Overview of the GIL framework.

### 2.1 PROBLEM FORMULATION

We consider a dataset containing $N$ observations $\mathcal{X} = (\mathbf{X}_1, \mathbf{X}_2, \ldots, \mathbf{X}_N)$ where each $\mathbf{X}_j$ can be represented by a vector $\mathbf{x}_j \in \mathbb{R}^d$ (for tabular data) or a matrix $\mathbf{X}_j \in \mathbb{R}^{T_j \times d}$ (for time-series) with

$T_j$ denoting the time horizon of the sequence $\mathbf{X}_j$. Since the focus of this work is principally on time-series data, recurrent neural networks will receive significant attention. We also define the set of missing indicators $\mathcal{M} = (\mathbf{M}_1, \mathbf{M}_2, \ldots, \mathbf{M}_N)$, where $\mathbf{M}_j = \mathbf{m}_j \in \{0, 1\}^d$ or $\mathbf{M}_j \in \{0, 1\}^{T_j \times d}$ depending on the dimension of $\mathbf{X}_j$, and 0's (or 1's) correspond to the entries in $\mathbf{X}_j$ that are missing (or not missing), respectively. Finally, we assume that each $\mathbf{X}_j$ is associated with some label $\mathbf{y}_j \in \mathcal{Y}$; thus $\mathcal{Y} = (\mathbf{y}_1, \mathbf{y}_2, \ldots, \mathbf{y}_N)$ denotes the set of labels. We define the problem as learning a model parameterized by $\mathbf{W}$ to directly predict $\mathbf{y}_j$ by generating $\hat{\mathbf{y}}_j$ that maximizes the log-likelihood $\sum_{j=1}^{N} \log p(\hat{\mathbf{y}}_j | \mathbf{X}_j, \mathbf{M}_j, \mathbf{W})$, without imputing the missing values in $\mathbf{X}_j$.

## 2.2 Gradient Importance

Missingness is a common issue found in tabular data and time series, where multi-layered perceptron (MLP) (Bishop, 2006) and long short-term memory (LSTM) (Hochreiter & Schmidhuber, 1997) models, respectively, are often considered for predictions. Below we illustrate how the *importance matrix* can be applied to gradients, which are produced by taking regular stochastic gradient descent (SGD) steps, toward training MLP models with incomplete tabular inputs. Note that this idea can be easily extended to sequential inputs with LSTM models and the corresponding details can be found in Appendix A. First, the definitions of the dataset and missing indicators can be reduced to $\mathcal{X}_{tab} = (\mathbf{x}_1, \mathbf{x}_2, \ldots, \mathbf{x}_N)$ and $\mathcal{M}_{tab} = (\mathbf{m}_1, \mathbf{m}_2, \ldots, \mathbf{m}_N)$, with $\mathbf{x}_j \in \mathbb{R}^d$ and $\mathbf{m}_j \in \mathbb{R}^d$, respectively. Then we define an MLP with $k$ hidden layers, for $k \geq 2$, as follows

$$\hat{\mathbf{y}} = \phi_{out}(\mathbf{W}_{out}\phi_k(\mathbf{W}_k \ldots \phi_2(\mathbf{W}_2\phi_1(\mathbf{W}_1\mathbf{x})))), \tag{1}$$

where $\mathbf{x} \in \mathbb{R}^d$ is the input, $\mathbf{W}_i$ and $\phi_i$ are the weight matrix and activation functions for the $i$-th hidden layer, respectively, and we have omitted the bias terms for notational convenience. The first layer can be interpreted as the *encoding* layer $\mathbf{W}_{enc} = \mathbf{W}_1$ that maps inputs to features $\zeta = f_{enc}(\mathbf{x}|\mathbf{W}_{enc})$, while the rest are the *inference* layers $\mathbf{W}_{inf} = \{\mathbf{W}_2, \ldots, \mathbf{W}_k, \mathbf{W}_{out}\}$ that map the features to prediction $\hat{\mathbf{y}} = f_{inf}(\zeta|\mathbf{W}_{inf})$. In the following proposition, we show that the gradients of some commonly used loss functions, $E(\hat{\mathbf{y}}, \mathbf{y})$, such as cross entropy or mean squared errors, w.r.t. $\mathbf{W}_1$, can be formulated in the form of an outer product. The proof and details can be found in Appendix E.

**Proposition 1** *Given a MLP and a smooth loss function $E(\hat{\mathbf{y}}, \mathbf{y})$, the gradients of $E$ w.r.t. the encoding layer can be formulated as $\partial E / \partial \mathbf{W}_1 = \boldsymbol{\Delta}_1 \mathbf{x}^\top$, where $\boldsymbol{\Delta}_1$ contains the gradients propagated from all the inference layers, $\mathbf{x} \in \mathcal{X}_{tab}$ and $\mathbf{y} \in \mathcal{Y}$.*

From this Proposition it can be seen that the gradients that are used to train the encoding layers, following regular SGD solvers, can be formulated as the *outer product* between the gradients $\boldsymbol{\Delta}$ propagated from the inference layers and the input $\mathbf{x}$ as

$$(\partial E / \partial \mathbf{W}_{enc})_{SGD} = \boldsymbol{\Delta} \cdot \mathbf{x}^\top, \tag{2}$$

where $\mathbf{W}_{enc} \in \mathbb{R}^{e \times d}$, $\boldsymbol{\Delta} \in \mathbb{R}^e$, $\mathbf{x} \in \mathbb{R}^d$, $e$ is the dimension of the features, and $d$ is the dimension of the input. To simplify notation, note that henceforth we use $\mathbf{x}$ to refer to an individual tabular data $\mathbf{x}_j \in \mathcal{X}_{tab}$, regardless of its index.

The corresponding SGD updates for training $\mathbf{W}_{enc}$, using learning rate $\alpha$, are $\mathbf{W}_{enc} \leftarrow \mathbf{W}_{enc} - \alpha\boldsymbol{\Delta} \cdot \mathbf{x}^\top$. As a result, it is observed from (2) that the $j$-th ($j \in [1, d]$) column of $(\partial E / \partial \mathbf{W}_{enc})_{SGD}$ is *weighted* by the $j$-th element in $\mathbf{x}$ given $\boldsymbol{\Delta}$. However, given that some entries in $\mathbf{x}$ could be *missing* (according to the missing indicator $\mathbf{m}$) and their values are usually replaced by a placeholder $\xi \in \mathbb{R}$, it may not be ideal do directly back-propagate the gradients in the form of (2), as the features $\zeta$ captured by the encoding layers may not be expressive enough for the inference layers to make accurate predictions. Instead, we consider introducing the *importance matrix* $\mathbf{A} \in [0, 1]^{e \times d}$ to adjust the gradient descent direction as

$$\mathbf{W}_{enc} \leftarrow \mathbf{W}_{enc} - \alpha(\partial E / \partial \mathbf{W}_{enc})_{SGD} \odot \mathbf{A}, \tag{3}$$

which not only trains the model to capture information from the observed inputs that are most useful for making accurate predictions, but also to leverage the information underlying the missingness in the data. The elements of $\mathbf{A}$ can be generated using the RL framework introduced in the next section.

Note that above we have omitted all the bias terms to simplify our presentation and note that: *i*) gradients w.r.t. to biases do not conform to the outer product format, and *ii*) more importantly, these gradients do not depend on the inputs – thus in practice, the importance matrix $\mathbf{A}$ is only applied

to the gradients of the weights. Moreover, we do not consider convolutional neural network (CNN) models (LeCun et al., 1999) in this work because of $i$). Though the proposed framework could still be applied to CNNs, it may not be as efficient as for MLPs or LSTMs, where the search space for $\mathbf{A}$ is significantly reduced by taking advantages of the outer product format, as shown in (4) below. Importantly, our focus is to address the missingness in tabular data and time-series, in which case MLPs and LSTMs are appropriate, while the missingness in images is usually related to applications in other domains such as compressed sensing (Wu et al., 2019) or image super-resolution (Dong et al., 2015), which we plan to explore in the future.

## 2.3 RL TO GENERATE IMPORTANCE MATRIX

Now we show how to use RL to generate the importance matrix $\mathbf{A}$. In general, RL aims to learn an optimal policy that maximizes expected rewards, by interacting with an unknown environment (Sutton & Barto, 2018). Formally, the RL environment is characterized by a Markov decision process (MDP), with details introduced below. Each time after the agent chooses an action $a = \pi(s)$ at state $s$ following some policy $\pi$, the environment responds with the next state $s'$, along with an immediate reward $r = R(s, a)$ given a reward function $R(\cdot, \cdot)$. The goal is to learn an optimal policy that maximizes the *expected total reward* defined as $J(\pi) = \mathbb{E}_{\rho_\pi}[\sum_{i=0}^{T} \gamma^i r(s_i, a_i)]$, where $T$ is the horizon of the MDP, $\gamma \in [0, 1)$ is the discounting factor, and $\rho_\pi = \{(s_0, a_0), (s_1, a_1), \ldots | a_i = \pi(s_i)\}$ is the sequence of states and actions drawn from the trajectory distribution determined by $\pi$.

In our case, the elements of $\mathbf{A}$ are determined on-the-fly following an RL policy $\pi_\theta$, parameterized by $\theta$, conditioned on some states that characterize the status of the back-propagation at the current time step. The policy $\pi_\theta$ is updated, concurrently with the weights $\mathbf{W}_{enc}$, by an RL agent that interacts with the back-propagation process modeled as the MDP defined as $\mathcal{M} = (\mathcal{S}, \mathcal{A}, \mathcal{P}, R, \gamma)$, with each of its elements introduced below.

**State Space** $\mathcal{S}$. The state space characterizes the agent's knowledge of the environment at each time step. To constitute the states we consider the 4-tuple $\mathbf{s} = (\mathbf{x}, \mathbf{m}, \zeta, \hat{\mathbf{y}})$ including current training input $\mathbf{x} \in \mathbb{R}^d$, the missing indicator $\mathbf{m} \in \mathbb{R}^d$, the feature $\zeta$, *i.e.*, the outputs of the encoding layer of MLPs or the hidden states $\mathbf{h}_t$ of LSTMs, and the predictions $\hat{\mathbf{y}}$ produced by the inference layers.

**Action Space** $\mathcal{A}$. We consider combining the *importance matrix* $\mathbf{A} \in [0, 1]^{e \times d}$ with the parameter gradients $(\partial E / \partial \mathbf{W}_{enc})_{SGD}$ when updating $\mathbf{W}_{enc}$ following

$$\mathbf{W}'_{enc} \leftarrow \mathbf{W}_{enc} - \alpha(\partial E / \partial \mathbf{W}_{enc})_{SGD} \odot \mathbf{A} = \mathbf{W}_{enc} - \alpha \mathbf{\Delta} \cdot (\mathbf{x}^\top \odot \mathbf{a}^\top); \qquad (4)$$

where here the equality follows from (2) and Proposition 1, such that all rows of $\mathbf{A}$ can be set to be equal to the *importance* $\mathbf{a}^\top \in [0, 1]^d$, which is obtained from the policy following $\mathbf{a} = \pi_\theta(\mathbf{s})$.

**Transitions** $\mathcal{P} : \mathcal{S} \times \mathcal{A} \to \mathcal{S}$. The transition dynamics determines how to transit from a current state $\mathbf{s}$ to the next state $\mathbf{s}'$ given an action $\mathbf{a}$ in the environment. In our case, the pair $(\mathbf{x}, \mathbf{m})$ is sampled from the training dataset at each step, so the transitions $\mathbf{x} \to \mathbf{x}'$ and $\mathbf{m} \to \mathbf{m}'$ are determined by how training samples are selected from the training batch during back-propagation. The update of $\mathbf{W}_{enc} \to \mathbf{W}'_{enc}$ is conditioned on the importance $\mathbf{a}$ as captured in (4), which results in the transitions $\zeta = f_{enc}(\mathbf{x}|\mathbf{W}_{enc}) \to \zeta' = f_{enc}(\mathbf{x}'|\mathbf{W}'_{enc})$. The update of $\mathbf{W}_{inf} \to \mathbf{W}'_{inf}$ follows from the regular SGD updates. Then, the transition $\hat{\mathbf{y}} = f(\mathbf{x}|\mathbf{W}) \to \hat{\mathbf{y}}' = f(\mathbf{x}'|\mathbf{W}')$ follows from the other transitions defined above. Finally, we can define the transition between states as $\mathbf{s} = (\mathbf{x}, \mathbf{m}, \zeta, \hat{\mathbf{y}}) \to \mathbf{s}' = (\mathbf{x}', \mathbf{m}', \zeta', \hat{\mathbf{y}}')$.

**Reward Function** $R$. After the RL agent takes an action $\mathbf{a}$ at the state $\mathbf{s}$, an immediate reward $r = R(\mathbf{s}, \mathbf{a})$ is returned by the environment which provides essential information to update $\pi_\theta$ (Silver et al., 2014; Lillicrap et al., 2015). We define the reward function as $R(\mathbf{s}, \mathbf{a}) = -E(\hat{\mathbf{y}}, \mathbf{y})$.

We utilize actor-critic RL (Silver et al., 2014; Lillicrap et al., 2015) to update $\pi_\theta$, which outputs the importance $\mathbf{a}$ that is used to concurrently update $\mathbf{W}_{enc}$. Specifically, we train the target policy (or actor) $\pi_\theta$ along with the critic $Q_\nu : \mathcal{S} \times \mathcal{A} \to \mathbb{R}$, parameterized by $\nu$, by maximizing $J_\beta(\pi_\theta) = \mathbb{E}_{s \sim \rho_\beta}[Q_\nu(s, \pi_\theta(s))]$ which gives an approximation to the expected total reward $J(\pi)$. Specifically, the trajectories $\rho_\beta = \{(s_0, a_0), (s_1, a_1), \ldots | a_i = \beta(s_i)\}$ collected under the behavioral policy $\beta : \mathcal{S} \to \mathcal{A}$ are used to update $\theta$ and $\nu$ jointly following

$$\nu' \leftarrow \nu + \alpha_\nu \delta \nabla_\nu Q_\nu(\mathbf{s}, \mathbf{a}), \quad \theta' \leftarrow \theta + \alpha_\theta \nabla_\theta \pi_\theta(\mathbf{s}) \nabla_a Q_\nu(\mathbf{s}, \mathbf{a})|_{\mathbf{a} = \pi_\theta(\mathbf{s})}; \qquad (5)$$

where $\beta$ is usually obtained by adding noise to the output of $\pi_\theta$ to ensure sufficient exploration of the state and action space, $\delta = r + \gamma Q_\nu(\mathbf{s}', \pi_\theta(\mathbf{s}')) - Q_\nu(\mathbf{s}, \mathbf{a})$ is the temporal difference error (residual) in RL, and $\alpha_\theta, \alpha_\nu$ are the learning rates for $\theta, \nu$, respectively.

---

**Algorithm 1** Gradient Importance Learning (GIL).

**Input:** $\mathcal{X}, \mathcal{Y}, \mathcal{M}, \mathbf{W}_{enc}, \mathbf{W}_{inf}, \pi_\theta, Q_\nu, \alpha_\theta, \alpha_\nu, \alpha, E$
**Begin:**
1: Initialize $\mathbf{W}_{enc}$ and $\mathbf{W}_{inf}$, actor $\pi_\theta$ and critic $Q_\nu$
2: Sample $\mathbf{x}$ from $\mathcal{X}$ and obtain the corresponding label $\mathbf{y}$ from $\mathcal{Y}$
3: Obtain the feature $\zeta \leftarrow f_{enc}(\mathbf{x}|\mathbf{W}_{enc})$ and prediction $\hat{\mathbf{y}} = f_{inf}(\zeta|\mathbf{W}_{inf})$ from the encoding and inference layers, respectively
4: $\mathbf{s} \leftarrow (\mathbf{x}, \mathbf{m}, \zeta, \hat{\mathbf{y}})$
5: **for** $iter$ in $1 : max\_iter$ **do**
6:     Obtain importance from a behavioral policy $\mathbf{a} = \beta(\mathbf{s}|\pi_\theta)$
7:     Train the encoding layer following $\mathbf{W}'_{enc} \leftarrow \mathbf{W}_{enc} - \alpha\mathbf{\Delta} \cdot (\mathbf{x}^\top \odot \mathbf{a}^\top)$ as in (4)
8:     Train the inference layers following regular gradient descent, *i.e.*,
        $\mathbf{W}'_{inf} \leftarrow \mathbf{W}_{inf} - \alpha(\partial E/\partial\mathbf{W}_{inf})_{SGD}$
9:     Obtain the prediction following the updated weights $\hat{\mathbf{y}} \leftarrow f(\mathbf{x}|\mathbf{W}'_{enc}, \mathbf{W}'_{inf})$
10:    Obtain the reward $r \leftarrow R(\mathbf{s}, \mathbf{a})$
11:    Get a new sample $\mathbf{x}'$ from $\mathcal{X}$ and obtain the corresponding label $\mathbf{y}'$ from $\mathcal{Y}$
12:    Obtain the feature $\zeta' \leftarrow f_{enc}(\mathbf{x}'|\mathbf{W}'_{enc})$ and prediction $\hat{\mathbf{y}}' = f_{inf}(\zeta'|\mathbf{W}'_{inf})$ from the encoding and inference layers, respectively
13:    $\mathbf{s}' \leftarrow (\mathbf{x}', \mathbf{m}', \zeta', \hat{\mathbf{y}}')$
14:    Update the actor $\pi_\theta$ and critic $Q_\nu$ using $(\mathbf{s}, \mathbf{a}, r, \mathbf{s}')$ following (5)
15:    $\mathbf{s} \leftarrow \mathbf{s}', \mathbf{W}_{enc} \leftarrow \mathbf{W}'_{enc}, \mathbf{W}_{inf} \leftarrow \mathbf{W}'_{inf}$
16: **end for**

---

We summarize the GIL approach in Algorithm 1 and the detailed descriptions can be found in Appendix B. Note that the missing indicator $\mathbf{m} \in \mathbb{R}^d$ would be a good heuristic to replace the importance $\mathbf{a}$ as it could prevent the gradients produced by the missing dimensions from propagation. However, it does not train the model to capture the hidden information underlying the missing data, which results in its performance being dominated by GIL as demonstrated in Section 4.

## 2.4 EXTENSIONS OF THE FRAMEWORK

The proposed GIL framework uses RL agents to guide the model toward minimizing its objective $E(\hat{\mathbf{y}}, \mathbf{y})$, which is usually captured by general prediction losses such as cross entropy or mean squared error. Below we use an example to show how our method can be extended to adapt ideas from related domains, which can augment training of the imputation-free prediction models by altering the reward function. Recent works in contrastive learning, *e.g.*, Chen et al. (2020), can train DL models to generate highly expressive features by penalizing (or promoting) the distributional difference $D(\zeta^+, \zeta^-)$ among the features associated with inputs that are similar to (or different from) each other, where $\zeta^+$ denotes the set of features corresponding to one set of homogeneous inputs $\mathcal{X}_{con}$, and $\zeta^-$ are the features generated by data that are significantly different from the ones in $\mathcal{X}_{con}$. However, such methods may not be directly applied to the missing data setting considered in this work, as their loss $D$ is usually designed toward unsupervised training and might need to be carefully re-worked in our case. This idea can be adapted to our framework by defining $\zeta^+$ as the features mapped from the inputs (by the encoding layers $\mathbf{W}_{enc}$) associated with the same label $y \in \mathcal{Y}$ and $\zeta^-$ the features corresponding to some other label $y' \in \mathcal{Y}$. Then we can define the new reward function $R(\mathbf{s}, \mathbf{a}) = -E(\hat{\mathbf{y}}, \mathbf{y}) + c \cdot D(\zeta^+, \zeta^-), c > 0$, which does not require $D$ to be carefully crafted, provided that $\mathbf{W}_{enc}$ is not trained by directly propagating gradients from it. In Section 4 it will be shown in case studies that by simply defining $D$ as the mean squared error between $\zeta^+$ and $\zeta^-$ can improve the prediction performance.

## 3 RELATED WORK

**Missing Data Imputation.**    Traditional mean/median-filling and carrying-forward imputation methods are still used widely, as they are straightforward to implement and interpret (Honaker & King, 2010). Recently, there have been state-of-the-art imputation algorithms (IAs) proposed to produce smooth imputations with interpretable uncertainty estimates. Specifically, some adopt Bayesian methods where the observed data are fit to data-generating models, including Gaussian processes (Wilson et al., 2016; Fortuin & Rätsch, 2019; Fortuin et al., 2018), multivariate imputation by chained equations (MICE) (Azur et al., 2011), random forests (Stekhoven & Bühlmann, 2012), *etc*., or statistical

optimization methods such as expectation-maximization (EM) (García-Laencina et al., 2010; Bashir & Wei, 2018). However, they often suffer from limited scalability, require assumptions over the underlying distribution, or fail to generalize well when used with datasets containing mixed types of variables, *i.e.*, when discrete and continuous values exist simultaneously (Yoon et al., 2018; Fortuin et al., 2020). There also exist matrix-completion methods that usually assume the data are static, *i.e.*, information does not change over time, and often rely on low-rank assumptions (Wang et al., 2006; Yu et al., 2016; Mazumder et al., 2010; Schnabel et al., 2016). More recently, several DL-based IAs have been proposed following advancements in deep generative models such as deep latent variable models (DLVMs) and generative adversarial networks (GANs) (Mattei & Frellsen, 2019; Yoon et al., 2018; Fortuin et al., 2020; Li et al., 2019; Ma et al., 2018). DLVMs are usually trained to capture the underlying distribution of the data, by maximizing an evidence lower bound (ELBO), which could introduce high variations to the output distributions (Kingma & Welling, 2013) and cause poor performance in practice, if the data have high missingness (as for the example shown in Figure 1). There also exist end-to-end methods that withhold observed values from inputs and impose reconstruction losses during training of prediction models (Cao et al., 2018; Ipsen et al., 2020). In addition, data-driven IAs are developed specifically toward medical time series by incorporating prior domain knowledge (Yang et al., 2018; Scherpf et al., 2019; Calvert et al., 2016; Gao et al., 2021). On the other hand, a few recent works attempt to address missing inputs through an imputation-free manner (Morvan et al., 2020; Sportisse et al., 2020), however, they are limited to linear regression.

**Attention.** The importance matrix used in the GIL is somewhat reminiscent of visual attention mechanisms in the context of image captioning or multi-object recognition (Mnih et al., 2014; Ba et al., 2014; Xu et al., 2015) as they both require training of the prediction models with RL. We briefly discuss the connections and distinctions between them, with full details provided in Appendix C. Visual attentions are commonly used to train CNNs to focus on specific portions of the inputs that are most helpful for making predictions. They are determined by maximizing an evidence lower bound (ELBO), which is later proved to be equivalent to the REINFORCE algorithm in the RL literature (Sutton & Barto, 2018; Mnih et al., 2014). However, these methods cannot be applied directly to our problem, as they require features to be exclusively associated *spatially* with specific parts of the inputs, which is attainable by using convolutional encoders with image inputs but intractable with the type of data considered in our work. Instead, our approach overcomes this issue by directly applying importance weights, generated by RL, into the *gradient space* during back-propagation. Such issues motivate use of the term *importance* instead of *attention*. Lastly, our method does not require one to formulate the learning objective as an ELBO, and as a result GIL can adopt any state-of-the-art RL algorithm, without being limited to REINFORCE as in Mnih et al. (2014); Ba et al. (2014); Xu et al. (2015). Besides, other types of attention mechanisms are also proposed toward applications in sequence modeling and natural language processing (NLP) such as Cheng et al. (2016); Vaswani et al. (2017).

## 4 EXPERIMENTS

We evaluate the performance of the *imputation-free* prediction models trained by GIL against the existing *imputation-prediction* paradigm on both benchmark and real-world datasets, where the imputation stage employs both state-of-the-art and classic imputation algorithms (IAs), with the details introduced in Section 4.1. We also consider variations of the GIL method proposed in Section 2 to illustrate its robustness and flexibility. The datasets we use include $i$) MIMIC-III (Johnson et al., 2016) that consists of real-world EHRs obtained from intensive care units (ICUs), $ii$) a de-identified ophthalmic patient dataset obtained from an eye center in North America, and $iii$) hand-written digits MNIST (LeCun & Cortes). We also tested on a smaller scaled ICU time-series from 2012 Physionet challenge (Silva et al., 2012) and these results can be found in Appendix D.4. Some of the data are augmented with additional missingness to ensure sufficient missing rates and the datasets we use cover all types of missing data, *i.e.,* missing complete at random (MCAR), MAR and MNAR (Little & Rubin, 2019). We found that the proposed method not only outperforms the baselines on all three datasets under various experimental settings, but also offers better feature representations as shown in Figure 3. We start with an overview of the baseline methods, in the following sub-section, and then proceed to present our experimental findings.

### 4.1 VARIANTS OF GIL AND BASELINES

Our method (GIL) trains MLPs or LSTMs (depending on the type of data) to directly output the predictions given incomplete inputs without imputation. Following from Section 2.4, we also test on

Table 1: Accuracy and AUC obtained from the MIMIC-III dataset.

|  |  | GIL | -D | -H | GAIN | MIWAE | GP-VAE | BRITS | MICE | Mean | CF | kNN | MF | EM |
|---|---|---|---|---|---|---|---|---|---|---|---|---|---|---|
| **Var-l.** | Acc. | **93.32** | 93.09 | 89.17 | 90.32 | 88.71 | - | - | 92.17 | 88.02 | 87.32 | 84.79 | 75.81 | 68.20 |
|  | AUC | **96.10** | **96.79** | 92.96 | 95.57 | 94.28 | - | - | 95.97 | 92.56 | 91.78 | 91.86 | 81.73 | 75.23 |
| **Fix-l.** | Acc. | **91.47** | 91.01 | 88.25 | 88.48 | 86.18 | 76.50 | 80.24 | 90.09 | 86.41 | 86.87 | 85.48 | 78.11 | 70.51 |
|  | AUC | **95.29** | 95.57 | 92.99 | 91.94 | 93.10 | 81.47 | 92.13 | 94.02 | 91.69 | 91.98 | 92.38 | 84.54 | 79.97 |

binary classification tasks using GIL-D which includes the distributional difference term $D(\zeta^+, \zeta^-)$, captured by mean squared errors, into the reward function. For baselines, we start with another variant of GIL that uses a simple heuristic – the missing indicator $\mathbf{m}$ – to replace the importance $\mathbf{a}$ in GIL; we denote it as GIL-H. This helps analyze the advantages of training the models using the importance obtained from the RL policy learned by GIL, *versus* a heuristic that simply discards the gradients produced by the subset of input entries that are missing.

For other baselines, following the imputation-prediction framework, the imputation stage employs state-of-the-art IAs including MIWAE (Mattei & Frellsen, 2019) which imputes the missing data by training variational auto-encoders (VAEs) to maximize an ELBO, GP-VAE (Fortuin et al., 2020) that couples VAEs with Gaussian processes (GPs) to consider the temporal correlation in time-series, and the generative adversarial network-based method GAIN (Yoon et al., 2018). Further, some classical imputation methods are also considered, including MICE, missForest (MF), $k$-nearest neighbor (kNN), expectation-maximization (EM), mean-imputation (Mean), zero-imputation (Zero) and carrying-forward (CF). The imputed data from these baselines are fed into the prediction models that share the same architecture as the ones trained by GIL, for fair comparison. For time-series, we also compare to BRITS (Cao et al., 2018) which trains bidirectional LSTMs end-to-end by masking out additional observed data and jointly imposing imputation and classification objectives.

## 4.2 MIMIC-III

We consider the early prediction of septic shock using the MIMIC-III dataset, following the frameworks provided in recent data-driven works for data pre-processing (Sheetrit et al., 2017; Fleuren et al., 2020; Khoshnevisan et al., 2020). Specifically, we use 14 commonly-utilized vital signs and lab results over a 2-hour observation window to predict whether a patient will develop septic shock in the next 4-hour window. The resulting dataset used for training and testing contains 2,166 sequences each of which has length between 3 and 125, and the overall missing rate is 71.63%; the missing data in this dataset can be considered as a mixture of MCAR, MAR and MNAR. More details can be found in Appendix D.1.

To evaluate performance, we consider the prediction model constituted by a 1024-unit LSTM layer for encoding followed by a dense output layer for inference. Considering that some of the baseline methods are not designed to capture the temporal correlations within the sequences which do not follow a fixed horizon, besides testing with the varied-length sequences (Var-l.) obtained in above, we also test with a fixed-length version (Fix-l.) where the maximum time step for each sequence is set to be 8 (see Appendix D.1 for details). The accuracy[1] and AUC for the two tests are summarized in Table 1. When varied-length sequences are considered, GIL(-D) slightly outperforms GAIN and MICE while it significantly dominates the other baseline methods. However, when fixed-length sequences are considered, GAIN and MICE's performance decreases dramatically and are significantly dominated by that of GIL(-D). This could be caused by omitting the records beyond the 8-th time step as both models are flexible enough to capture such information from that point onwards, while in contrast, the performance increases for kNN, MF and EM which are in general based on much simpler models. On the other hand, the models trained by GIL(-D) was not significantly affected by the information loss. In fact, it emphasizes that applying importance to the gradients during training can enable the model to capture the information behind missing data. Moreover, the dramatically increased performance from GIL-H to GIL(-D) in both tests underscores the significance of training the models using the importance determined by the RL policy learned by GIL(-D), instead of a pre-defined heuristic. Note that according to a recent survey of septic shock predictions (Fleuren et al., 2020), the overall maximum AUC attained by existing data-driven methods, which uses domain knowledge to design models specifically for ICU time-series, is 0.96 and it is comparable to that attained by our method. Finally, GP-VAE and BRITS require the input sequences to have the same horizon, so we only report for the Fix-l. case where they under-perform. These are possibly due to

---

[1]All accuracy values in this work are obtained using a decision threshold of 0.5.

Table 2: Average Accuracy and AUC obtained from the Ophthalmic dataset over 3 different random masks. Subscripts are standard deviations.

| | M.R. | GIL | -D | -H | GAIN | GP-VAE | MIWAE | MICE | Zero |
|---|---|---|---|---|---|---|---|---|---|
| Acc. | 25% | $86.84_{1.43}$ | $87.13_{1.09}$ | $83.63_{0.83}$ | $85.38_{1.09}$ | $85.67_{0.83}$ | $80.99_{1.8}$ | $84.21_{0.72}$ | $84.50_{1.8}$ |
| AUC | | $92.40_{1.33}$ | $92.42_{1.44}$ | $88.87_{2.16}$ | $90.13_{2.09}$ | $\underline{91.47}_{1.02}$ | $84.04_{1.47}$ | $91.35_{1.35}$ | $90.59_{0.32}$ |
| Acc. | 35% | $83.33_{0.72}$ | $85.09_{2.58}$ | $80.41_{1.49}$ | $80.41_{0.83}$ | $80.41_{0.83}$ | $79.24_{4.38}$ | $80.41_{1.49}$ | $80.12_{1.09}$ |
| AUC | | $88.49_{0.68}$ | $90.68_{2.36}$ | $87.02_{3.03}$ | $\underline{88.02}_{2.15}$ | $84.85_{3.29}$ | $85.51_{3.47}$ | $85.87_{3.24}$ | $\underline{88.02}_{0.97}$ |

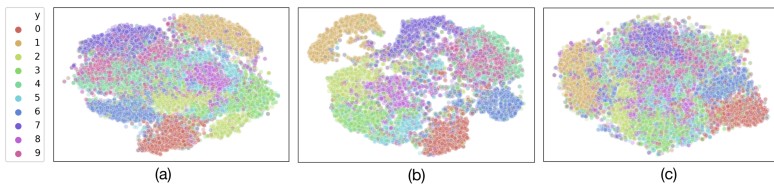

Figure 3: $t$-SNE visualization of the feature space learned by (a) GIL, (b) MIWAE and (c) GAIN on the MNIST dataset with 90% missing rate.

the high variation in GPVAE's uncertainty estimation component, and the mechanism of imposing additional missingness in BRITS[2].

### 4.3 OPHTHALMIC DATA

We consider identifying diabetic retinopathy (DR), an eye disease caused by diabetes, using the de-identified data obtained from Duke Eye Center, constituted by a mixture of image feature vectors and tabular information. Specifically, the data collected from each subject is formulated as a vector with 4,101 dimensions containing two 2,048-length feature vectors from two retinal imaging modalities, optic coherence tomography (OCT) and color fundus photography (CFP), followed by a 5-dimensional data characterizing demographic (*e.g.*, age) and diabetic information (*e.g.*, years of diabetes). Additional details of the dataset and examples of the OCT and CFP images are provided in the Appendix D.2. At most one of the two types of retinal images could be missing due to MAR as sometimes the diagnosis can be performed with a single modality (Cui et al., 2021), while the demographic/diabetic information can be missing due to MCAR. A total of 1,148 subjects are included in this dataset with a 17% missing rate (M.R.)[3]. We apply additional random masks over both the input image features (by assuming one or two entire image feature vectors are missing) and demographic/diabetic information (by assuming some entries are missing) to increase the missing rate to 25% and 35%, where for each case 3 different random masks are applied.

For this experiment we consider an MLP with 2 hidden layers with 1,000 nodes each for prediction. The results are shown in Table 2 where the mean and standard deviations (in subscripts) of accuracy and AUC are reported. Although GIL and GP-VAE result in similar performance in the 25% missing rate case, GIL still achieves higher accuracy and AUC; these are significantly improved, with significantly lower standard deviation over GP-VAE, when the missing rate increases to 35%. In general, GIL-D outperforms all the other methods, with an exception of the higher standard deviation of accuracy in the 35% M.R. case, which benefits from the flexibility of the framework we proposed. With increased missingness, the performance of most baselines drops significantly and are close to the zero-imputation baseline as the image feature vectors occupy most dimensions of the inputs while imputing with zeros would be a reasonable heuristic.

### 4.4 MNIST

This work focuses mostly on tabular inputs or time-series; however, we test on MNIST images given its simple structure, which could be classified using MLPs and more importantly, because results from these data are easier to interpret. Specifically, we test on the MCAR version of MNIST where a pre-determined portion of pixels (*i.e.*, out of 50%, 70% and 90%) are masked off uniformly at random from the original images, and the MAR version, which assumes part of the image is always observable and the positions of missing pixels are determined by a distribution conditioned on features extracted from the observable portion following Mattei & Frellsen (2019). For each test, the masks are applied

---

[2]The hyper-parameters used to train their models can be found in Appendix D.1

[3]When calculating missing rates, each feature vector is only counted as a single entry regardless of size.

Table 3: Average Accuracy reported for the MNIST dataset over different missing rates. For each missing rate, 5 random masks were used resulting in standard deviations shown as subscripts ($\times 10^{-3}$).

| | M.R. | GIL | -H | GAIN | MIWAE | MICE | Zero | GP-VAE |
|---|---|---|---|---|---|---|---|---|
| **MCAR** | 50% | $\mathbf{96.29}_{0.7}$ | $96.08_{1.2}$ | $96.23_{0.7}$ | $95.46_{0.8}$ | $94.58_{0.2}$ | $95.83_{1.7}$ | $96.57_{3.1}$ |
| | 70% | $\mathbf{93.35}_{0.9}$ | $93.26_{0.9}$ | $91.98_{3.8}$ | $93.20_{0.8}$ | $90.53_{3}$ | $92.80_{0.8}$ | $93.49_{3}$ |
| | 90% | $\mathbf{78.47}_{3.9}$ | $78.44_{4.7}$ | $72.58_{9}$ | $77.91_{11.3}$ | $73.48_{5}$ | $76.67_{5.3}$ | $76.25_{2}$ |
| **MAR** | - | $\mathbf{93.23}_{0.3}$ | $93.15_{0.5}$ | $92.97_{0.8}$ | $93.18_{0.4}$ | $92.62_{1.1}$ | $82.50_{2.9}$ | $92.62_{1.3}$ |

Table 4: Correlation between imputation MSE and prediction accuracy for different MRs.

| | MNIST | | | Ophthalmic | | MIMIC |
|---|---|---|---|---|---|---|
| | 50% M.R. | 70% M.R. | 90% M.R. | 25% M.R. | 35% M.R. | Ground-truths Unknown |
| Pearson's Coeff. | -0.48 | -0.88 | -0.86 | -0.12 | -0.54 | - |
| $p$-value | 0.018 | <0.01 | <0.01 | 0.581 | <0.01 | - |

with 5 different random seeds, resulting in the standard deviations reported in Table 3 which also summarizes obtained results.

We consider using MLPs constituted by two hidden dense layers with 500 nodes each as the prediction models[4]. Note that in the MCAR setting, GP-VAE achieves slightly higher average accuracy than our method when 50% and 70% of the pixels missing, because it relies on convolutional encoding layers, which intrinsically have advantages on this dataset. However, GP-VAE's performance is associated with higher standard deviation in both cases and is significantly outperformed by our method when 90% pixels are missing. The models trained by GIL also significantly outperforms the baselines following the imputation-prediction framework. This suggests that most of the IAs fail to reconstruct the input data when the missing rate is high (see Figure 1). Moreover, the errors produced during the imputation are then propagated into the prediction step which results in the inferior performance. In the MAR setting, the task becomes more challenging as entire rows of pixels could be missing. Finally, the model trained by GIL outperforms most of the baselines depending on IAs with an exception of MIWAE, which only slightly outperforms it. Note that GIL-H's performance is close to GIL in most of the settings due to the simple structure of the MNIST digits where the missing indicator $\mathbf{m}$ could be a good estimation of the importance $\mathbf{a}$.

**Feature Space Visualization** In Figure 3, we use $t$-distributed stochastic neighbor embedding ($t$-SNE) to visualize the features learned by the prediction model trained by GIL compared to MIWAE and GAIN. We observe that GIL results in more expressive features than the others as it generates clusters with clearer boundaries.

### 4.5 Correlation between Imputation and Prediction Performance

In this section we study if imputation error is correlated with the downstream prediction performance, under the imputation-prediction framework. Each column of Table 4 is obtained by calculating the Pearson's correlation coefficient (Freedman et al., 2007) and its $p$-value, between the imputation mean squared error (MSE) and prediction accuracy, across different imputation methods under the same dataset. It can be observed that the imputation MSE is negatively correlated with prediction performance for most of the datasets and data with higher M.R. tends to have higher degree of negative correlation, which indicates that imputation error could be propagated to downstream tasks.

### 5 Conclusion

We have developed the GIL method, training DL models (MLPs and LSTMs) to directly perform inference with incomplete data, without the need for imputation. Existing methods addressing the problem of missing data mostly follow the two-step (imputation-prediction) framework. However, the error produced during imputation can be propagated into subsequent tasks, especially when the data have high missingness rates or small sample sizes. GIL circumvents these issues by applying importance weighting to the gradients to leverage the information underlying the missingness patterns in the data. We have evaluated our method by comparing it to the state-of-the-art baselines using IAs on two real-world datasets and one benchmark dataset.

---

[4]The performance of MIWAE reported here is different from Mattei & Frellsen (2019) because they used a CNN classifier.

ACKNOWLEDGMENTS

This research was supported in part by the ONR under agreements N00014-17-1-2504 and N00014-20-1-2745, AFOSR under award number FA9550-19-1-0169, NIH under NIH/NIDDK R01-DK123062 and NIH/NINDS 1R61NS120246 awards, DARPA, as well as the NSF under CNS-1652544 award and the National AI Institute for Edge Computing Leveraging Next Generation Wireless Networks, Grant CNS-2112562.

We would also like to thank Ruiyi Zhang (Adobe Research) for advice that leads to improved paper presentations, and Ge Gao (North Carolina State University) for sharing insights toward processing the MIMIC-III dataset as well as the use of some imputation baselines.

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

# A    EXTENDING GIL TO LSTMS

**LSTM Model.**    We also consider the use of an LSTM as the *encoder* for sequential inputs with varied lengths. Specifically, in this case, we can define $\mathbf{X}_j = (\mathbf{x}_{j,1}^\top, \mathbf{x}_{j,2}^\top, \ldots, \mathbf{x}_{j,T_j}^\top)^\top \in \mathbb{R}^{T_j \times d}$ along with $\mathbf{M}_j = (\mathbf{m}_{j,1}^\top, \mathbf{m}_{j,2}^\top, \ldots, \mathbf{m}_{j,T_i}^\top) \in \{0,1\}^{T_j \times d}$ such that each $\mathbf{x}_{j,t} \in \mathbb{R}^d$ and $\mathbf{m}_{j,t} \in \{0,1\}^d$, where $i \in [1, N]$ and $t \in [1, T_j]$. Recall that the forward pass of an LSTM follows

$$\begin{aligned}
\mathbf{o}_t &= \sigma(\mathbf{W_o}\mathbf{x}_{j,t} + \mathbf{U_o}\mathbf{h}_{-1}), \ \ \mathbf{i}_t = \sigma(\mathbf{W_i}\mathbf{x}_{j,t} + \mathbf{U_i}\mathbf{h}_{-1}), \ \ \mathbf{f}_t = \sigma(\mathbf{W}_f\mathbf{x}_{j,t} + \mathbf{U_f}\mathbf{h}_{-1}), \\
\mathbf{g}_t &= \tanh(\mathbf{W_g}\mathbf{x}_{j,t} + \mathbf{U_g}\mathbf{h}_{-1}), \quad \mathbf{c}_t = \mathbf{f}_t \odot \mathbf{c}_{-1} + \mathbf{i}_t \odot \mathbf{g}_t, \quad \mathbf{h}_t = \mathbf{o}_t \odot \tanh(\mathbf{c}_t),
\end{aligned} \tag{6}$$

where $\mathbf{x}_t \in \mathbb{R}^d$ is the input at time step $t$, $\{\mathbf{W_o}, \mathbf{W_i}, \mathbf{W_g}, \mathbf{W_f}, \mathbf{U_o}, \mathbf{U_i}, \mathbf{U_g}, \mathbf{U_f}\}$ are the weights, $\sigma(x) = 1/(1 + e^{-x})$ is the sigmoid function, and $\odot$ is the element-wise product. The output layer is appended after $\mathbf{h}_t$ to constitute the *inference* layer, *i.e.*,

$$\hat{\mathbf{y}}_t = \phi_{out}(\mathbf{W}_{out}\mathbf{h}_t), \tag{7}$$

where $\phi_{out}$ is the activation function, $\mathbf{W}_{inf} = \mathbf{W}_{out}$ are the weights and $t \in [1, T_j]$. The following proposition shows that given a loss function $E(\hat{\mathbf{y}}, \mathbf{y})$, the gradients of $E$ w.r.t the encoding weights $\mathbf{W}_{enc} = \{\mathbf{W_o}, \mathbf{W_i}, \mathbf{W_g}, \mathbf{W_f}\}$ can be written in the form of outer products. Note that the gradients for (autoregressive) parameters depending on previous hidden states $\mathbf{h}_{t-1}$, *i.e.,*, $\{\mathbf{U_o}, \mathbf{U_i}, \mathbf{U_g}, \mathbf{U_f}\}$, like the bias terms, cannot be written as outer products; thus, these weights are updated following the regular SGD, without importance weighting. The proof is provided following the proposition.

**Proposition 2** *Given an LSTM as described in (6)-(7) and a smooth loss function $E(\hat{\mathbf{y}}, \mathbf{y})$, the gradients of $E$ w.r.t. $\mathbf{W}_{enc} = \{\mathbf{W_o}, \mathbf{W_i}, \mathbf{W_g}, \mathbf{W_f}\}$ at time $t$ can be written in outer product forms – i.e., it holds that $\frac{\partial E}{\partial \mathbf{W_o}}\big|_t = \boldsymbol{\Delta}_t^{\mathbf{o}}\mathbf{x}_t^\top$, $\frac{\partial E}{\partial \mathbf{W_i}}\big|_t = \boldsymbol{\Delta}_t^{\mathbf{i}}\mathbf{x}_t^\top$, $\frac{\partial E}{\partial \mathbf{W_g}}\big|_t = \boldsymbol{\Delta}_t^{\mathbf{g}}\mathbf{x}_t^\top$, $\frac{\partial E}{\partial \mathbf{W_f}}\big|_t = \boldsymbol{\Delta}_t^{\mathbf{f}}\mathbf{x}_t^\top$, where $\boldsymbol{\Delta}_t^{\mathbf{o}}, \boldsymbol{\Delta}_t^{\mathbf{i}}, \boldsymbol{\Delta}_t^{\mathbf{g}}, \boldsymbol{\Delta}_t^{\mathbf{f}}$ are the gradients propagated from the inference layers defined in below, and $\mathbf{x}_t$ represents the observation obtained at time step $t$ from any sequence $\mathbf{X}_j \in \mathcal{X}$ in general.*

Then it follows that if we define $\mathbf{x}$ as the $t$-th observation from any sequence $\mathbf{X}_j \in \mathbb{R}^{T_j \times d}$, regardless of it index, (4) can be directly used for training the LSTM encoding weights $\mathbf{W}_{enc} = \{\mathbf{W_o}, \mathbf{W_i}, \mathbf{W_g}, \mathbf{W_f}\}$.

Now we prove the Proposition above.

We first define

$$\boldsymbol{\Delta}_t^{\mathbf{o}} = \partial E/\partial \mathbf{h}_t \odot \tanh(\mathbf{c}_t) \odot \mathbf{o}_t \odot (1 - \mathbf{o}_t), \tag{8}$$

$$\boldsymbol{\Delta}_t^{\mathbf{i}} = \partial E/\partial \mathbf{h}_t \odot \mathbf{o}_t \odot (1 - \tanh^2(\mathbf{c}_t)) \odot \mathbf{g}_t \odot \mathbf{i}_t \odot (1 - \mathbf{i}_t), \tag{9}$$

$$\boldsymbol{\Delta}_t^{\mathbf{g}} = \partial E/\partial \mathbf{h}_t \odot \mathbf{o}_t \odot (1 - \tanh^2(\mathbf{c}_t)) \odot \mathbf{i}_t \odot (1 - \mathbf{g}_t^2), \tag{10}$$

$$\boldsymbol{\Delta}_t^{\mathbf{f}} = \partial E/\partial \mathbf{h}_t \odot \mathbf{o}_t \odot (1 - \tanh^2(\mathbf{c}_t)) \odot \mathbf{c}_{t-1} \odot \mathbf{f}_t \odot (1 - \mathbf{f}_t). \tag{11}$$

Now to prove the proposition we start from the derivative of the smooth loss function $E$ w.r.t. $\mathbf{x}$, which can be derived as

$$\frac{\partial E}{\partial \mathbf{h}_t} = \left[\frac{\partial(\mathbf{W}_{out}\mathbf{h}_t)}{\partial \mathbf{h}_t}\right]^\top. \tag{12}$$

$$\left[\frac{\partial E}{\partial \hat{\mathbf{y}}_t} \odot \phi'(\mathbf{W}_{out}\mathbf{h}_t)\right] \tag{13}$$

$$= \mathbf{W}_{out}^\top \left[\frac{\partial E}{\partial \hat{\mathbf{y}}_t} \odot \phi'(\mathbf{W}_{out}\mathbf{h}_t)\right]. \tag{14}$$

Then the derivatives of $E$ w.r.t. the $\mathbf{o}_t$, $\mathbf{c}_t$, $\mathbf{f}_t$, $\mathbf{c}_{t-1}$, $\mathbf{i}_t$ and $\mathbf{g}_t$ are

$$\frac{\partial E}{\partial \mathbf{o}_t} = \frac{\partial E}{\partial \mathbf{h}_t}\frac{\partial \mathbf{h}_t}{\partial \mathbf{o}_t}$$

$$= \frac{\partial E}{\partial \mathbf{h}_t} \odot \tanh(\mathbf{c}_t) \tag{15}$$

$$\frac{\partial E}{\partial \mathbf{c}_t} = \frac{\partial E}{\partial \mathbf{h}_t}\frac{\partial \mathbf{h}_t}{\partial \mathbf{c}_t}$$

$$= \frac{\partial E}{\partial \mathbf{h}_t} \odot \mathbf{o}_t \odot (1 - \tanh^2(\mathbf{c}_t)) \tag{16}$$

$$\frac{\partial E}{\partial \mathbf{f}_t} = \frac{\partial E}{\partial \mathbf{h}_t}\frac{\partial \mathbf{h}_t}{\partial \mathbf{c}_t}\frac{\partial \mathbf{c}_t}{\partial \mathbf{f}_t}$$

$$= \frac{\partial E}{\partial \mathbf{h}_t} \odot \mathbf{o}_t \odot (1 - \tanh^2(\mathbf{c}_t)) \odot \mathbf{c}_{t-1} \tag{17}$$

$$\frac{\partial E}{\partial \mathbf{c}_{t-1}} = \frac{\partial E}{\partial \mathbf{h}_t}\frac{\partial \mathbf{h}_t}{\partial \mathbf{c}_t}\frac{\partial \mathbf{c}_t}{\partial \mathbf{c}_{t-1}}$$

$$= \frac{\partial E}{\partial \mathbf{h}_t} \odot \mathbf{o}_t \odot (1 - \tanh^2(\mathbf{c}_t)) \odot \mathbf{f}_t \tag{18}$$

$$\frac{\partial E}{\partial \mathbf{i}_t} = \frac{\partial E}{\partial \mathbf{h}_t}\frac{\partial \mathbf{h}_t}{\partial \mathbf{c}_t}\frac{\partial \mathbf{c}_t}{\partial \mathbf{i}_t}$$

$$= \frac{\partial E}{\partial \mathbf{h}_t} \odot \mathbf{o}_t \odot (1 - \tanh^2(\mathbf{c}_t)) \odot \mathbf{g}_t \tag{19}$$

$$\frac{\partial E}{\partial \mathbf{g}_t} = \frac{\partial E}{\partial \mathbf{h}_t}\frac{\partial \mathbf{h}_t}{\partial \mathbf{c}_t}\frac{\partial \mathbf{c}_t}{\partial \mathbf{g}_t}$$

$$= \frac{\partial E}{\partial \mathbf{h}_t} \odot \mathbf{o}_t \odot (1 - \tanh^2(\mathbf{c}_t)) \odot \mathbf{i}_t. \tag{20}$$

Now, the derivatives of $E$ w.r.t the weights $\mathbf{W_o}$, $\mathbf{W_i}$, $\mathbf{W_g}$ and $\mathbf{W_f}$ are

$$\frac{\partial E}{\partial \mathbf{W_o}}\Big|_t = \frac{\partial E}{\partial \mathbf{h}_t}\frac{\partial \mathbf{h}_t}{\partial \mathbf{o}_t}\frac{\partial \mathbf{o}_t}{\partial \mathbf{W_o}} \tag{21}$$

$$= \left[\frac{\partial E}{\partial \mathbf{h}_t} \odot \tanh(\mathbf{c}_t) \odot \mathbf{o}_t \odot (1 - \mathbf{o}_t)\right]\mathbf{x}_t^\top \tag{22}$$

$$= \mathbf{\Delta}_t^{\mathbf{o}}\mathbf{x}_t^\top \tag{23}$$

$$\frac{\partial E}{\partial \mathbf{W_i}}\Big|_t = \frac{\partial E}{\partial \mathbf{h}_t}\frac{\partial \mathbf{h}_t}{\partial \mathbf{c}_t}\frac{\partial \mathbf{c}_t}{\partial \mathbf{i}_t}\frac{\partial \mathbf{i}_t}{\partial \mathbf{W_i}} \tag{24}$$

$$= \left[\frac{\partial E}{\partial \mathbf{h}_t} \odot \mathbf{o}_t \odot (1 - \tanh^2(\mathbf{c}_t)) \odot \mathbf{g}_t \odot \mathbf{i}_t \odot (1 - \mathbf{i}_t)\right]\mathbf{x}_t^\top \tag{25}$$

$$= \mathbf{\Delta}_t^{\mathbf{i}}\mathbf{x}_t^\top \tag{26}$$

$$\frac{\partial E}{\partial \mathbf{W_g}}\Big|_t = \frac{\partial E}{\partial \mathbf{h}_t}\frac{\partial \mathbf{h}_t}{\partial \mathbf{c}_t}\frac{\partial \mathbf{c}_t}{\partial \mathbf{g}_t}\frac{\partial \mathbf{g}_t}{\partial \mathbf{W_g}} \tag{27}$$

$$= \left[\frac{\partial E}{\partial \mathbf{h}_t} \odot \mathbf{o}_t \odot (1 - \tanh^2(\mathbf{c}_t)) \odot \mathbf{i}_t \odot (1 - \mathbf{g}_t^2)\right]\mathbf{x}_t^\top \tag{28}$$

$$= \mathbf{\Delta}_t^{\mathbf{g}}\mathbf{x}_t^\top \tag{29}$$

$$\frac{\partial E}{\partial \mathbf{W_f}}\Big|_t = \frac{\partial E}{\partial \mathbf{h}_t}\frac{\partial \mathbf{h}_t}{\partial \mathbf{c}_t}\frac{\partial \mathbf{c}_t}{\partial \mathbf{f}_t}\frac{\partial \mathbf{f}_t}{\partial \mathbf{W_f}} \tag{30}$$

$$= \left[\frac{\partial E}{\partial \mathbf{h}_t} \odot \mathbf{o}_t \odot (1 - \tanh^2(\mathbf{c}_t)) \odot \mathbf{c}_{t-1} \odot \mathbf{f}_t \odot (1 - \mathbf{f}_t)\right]\mathbf{x}_t^\top \tag{31}$$

$$= \mathbf{\Delta}_t^{\mathbf{f}}\mathbf{x}_t^\top. \tag{32}$$

Note that since $\sigma'(\cdot) = \sigma(\cdot)(1 - \sigma(\cdot))$, in the transition between (21) and (22) it follows that

$$\frac{\partial \mathbf{o}_t}{\partial \mathbf{W_o}} = \sigma'(\mathbf{W_o}\mathbf{x}_t + \mathbf{U_o}\mathbf{h}_{t-1})\mathbf{x}_t^\top \tag{33}$$

$$= \left[\sigma(\mathbf{W_o}\mathbf{x}_t + \mathbf{U_o}\mathbf{h}_{t-1}) \odot (1 - \sigma(\mathbf{W_o}\mathbf{x}_t + \mathbf{U_o}\mathbf{h}_{t-1}))\right]\mathbf{x}_t^\top \tag{34}$$

$$= \left[\mathbf{o}_t \odot (1 - \mathbf{o}_t)\right]\mathbf{x}_t^\top. \tag{35}$$

Similarly, the transition between (24) and (25) follows

$$\frac{\partial \mathbf{i}_t}{\partial \mathbf{W_i}} = \mathbf{i}_t \odot (1 - \mathbf{i}_t). \tag{36}$$

At last, the transition between (30) and (31) follows

$$\frac{\partial \mathbf{f}_t}{\partial \mathbf{W_f}} = \mathbf{f}_t \odot (1 - \mathbf{f}_t). \tag{37}$$

Furthemore, since $\tanh'(\cdot) = 1 - \tanh^2(\cdot)$, the transition between (27) and (28) follows

$$\frac{\partial \mathbf{g}_t}{\partial \mathbf{W_g}} = \tanh'(\mathbf{W_g}\mathbf{x}_t + \mathbf{U_g}\mathbf{h}_{t-1})\mathbf{x}_t^\top \tag{38}$$

$$= (1 - \tanh^2(\mathbf{W_g}\mathbf{x}_t + \mathbf{U_g}\mathbf{h}_{t-1}))\mathbf{x}_t^\top \tag{39}$$

$$= (1 - \mathbf{g}_t^2)\mathbf{x}_t^\top, \tag{40}$$

which concludes the proof.

## B  DESCRIPTION OF ALGORITHM 1

The algorithm takes as input the training dataset $\mathcal{X}$, the training targets $\mathcal{Y}$, the missing indicators $\mathcal{M}$, the weights of the encoding layers $\mathbf{W}_{enc}$, the weights for the inference layers $\mathbf{W}_{inf}$, the actor $\pi_\theta$, the critic $Q_\nu$, learning rates $\{\alpha, \alpha_\theta, \alpha_\nu\}$ and training loss function $E$. Our approach starts by initializing all the parameters $\mathbf{W}_{enc}$, $\mathbf{W}_{inf}$, $\pi_\theta$, $Q_\nu$, sampling $\mathbf{x} \in \mathcal{X}$ with the corresponding $\mathbf{m} \in \mathcal{M}$ that will be used for training in the first iteration, obtaining the feature $\zeta$ and the prediction $\hat{\mathbf{y}}$, which constitute the initial state as $\mathbf{s} = (\mathbf{x}, \mathbf{m}, \zeta, \hat{\mathbf{y}})$. In each iteration, first the importance is generated from a behavioral policy $\beta$ that is conditioned on the target policy $\pi_\theta$, such as the noisy exploration policy proposed in Lillicrap et al. (2015). Then the encoding layer is trained following (4), while the inference layers are trained following the regular gradient descent. After training, the new prediction is obtained following the *updated* weights and its value is assigned to $\hat{\mathbf{y}}$, which is then used to generate the reward following the reward function $R$. Then the training sample $\mathbf{x}'$ for the next iteration is sampled and the corresponding $\mathbf{m}', \zeta', \hat{\mathbf{y}}'$ are obtained to constitute the next state $\mathbf{s}'$. Finally, the actor $\pi_\theta$ and critic $Q_\nu$ are updated following (5). We refer to Silver et al. (2014); Lillicrap et al. (2015) for more details on actor-critic RL.

## C  IMPORTANCE VS ATTENTIONS

We now illustrate the connections and distinctions between the importance in the GIL and visual attentions Mnih et al. (2014); Ba et al. (2014); Xu et al. (2015), which are commonly used to train CNNs to focus on specific dimensions of the inputs that are most helpful for making predictions in the context of image captioning and multi-object recognition. In visual attentions, an input image is first encoded into a set of vectors $\mathcal{I} = \{\mathbf{l}_1, \ldots, \mathbf{l}_L\}$ where each $\mathbf{l}_i$ is a feature vector corresponding to *a specific region* in the image as they are retrieved from lower convolutional layers. Then, the attention $\alpha_{t,i}$ for time step $t$ and region $i \in [1, L]$ is generated following Bahdanau et al. (2014) as $\alpha_{t,i} = \exp(e_{t,i}) / \sum_{j=1}^{L} \exp(e_{t,k})$, where $e_{t,i} = f_{att}(\mathbf{l}_i, \mathbf{h}_{t-1})$, $f_{att}$ is usually an MLP (or the so-called attention model) and $\mathbf{h}_{t-1}$ is the hidden state of a recurrent neural network (RNN) Hochreiter & Schmidhuber (1997) used for prediction. Then, a weighted average of the feature vectors $\sum_i l_{t,i} \mathbf{l}_i$ is fed into the prediction network for further inference where $l_{t,i}$ is a random variable parametrized by $\alpha_{t,i}$ following $p(l_{t,i} = 1 | l_{j<t}, \mathcal{I}) = \alpha_{t,i}$, and $l_{j<t}$ represents the historical values of $l_{t,i}$ for all $i \in [1, L]$.

Given the definition of the multinoulli variable $l_{t,i}$, the model is not smooth and thus cannot be trained following the regular back-propagation. So the prediction network is trained to maximize the evidence lower bound (ELBO) which updates the prediction network parameter $\theta$ using

$$\frac{\partial J}{\partial \theta} \approx \frac{1}{n} \sum_{i=1}^{n} \left[ \frac{\partial \log p(\mathbf{y}|\tilde{l}_i, \mathcal{I})}{\partial \theta} + (\log p(\mathbf{y}|\tilde{l}_i, \mathcal{I}) - b) \frac{\partial \log p(\tilde{l}_i|\mathcal{I})}{\partial \theta} \right], \tag{41}$$

where $\tilde{l}_i$ are the Monte-Carlo samples drawn from $p(l_{t,i}|l_{j<t}, \mathcal{I})$ and $b$ represents the performance from a baseline $\mathbb{E}[\log p(\mathbf{y}|\theta_{baseline})]$. It is notable that (41) is equivalent to the REINFOCE algorithm in RL Williams (1992). Specifically, the search space can be interpreted as an MDP which is usually used to model the environment in RL problems – i.e., the state space is constituted by $\mathcal{I}$, the RNN hidden state $\mathbf{h}_t$ and the historical visitations $l_{j<t}$; the actions are multinouli variables $l_{t,i}$; the reward function is defined as the marginal log-likelihood $\log p(\mathbf{y}|l_{t,i}, \mathcal{I})$; and the policy is characterized by $p(l_{t,i}|l_{j<t}, \mathcal{I})$ Xu et al. (2015); Mnih et al. (2014); Ba et al. (2014).

However, these methods cannot be applied directly to our problem as it requires the features $\mathcal{I}$ to exclusively associate with specific parts of the inputs *spatially*, which is attainable by using convolutional encoders with image inputs but intractable with tabular inputs considered in our case (see Section 2.1). Instead, our approach overcomes this issue by directly applying importance, generated by RL, into the *gradient space* during back-propagation. Moreover, our method does not require to formulate the learning objective as ELBOs and as a result GIL can adopt any state-of-the-art RL algorithm – not limited to REINFORCE only as in Mnih et al. (2014); Ba et al. (2014); Xu et al. (2015).

# D    EXPERIMENTAL DETAILS AND ADDITIONAL EXPERIMENTS

The case studies are run on a work station with three Nvidia Quadro RTX 6000 GPUs with 24GB of memory for each. We use Tensorflow to implement the models and training algorithms. To train the imputation-free prediction models using GIL, we perform a grid search for the model learning rate $\alpha \in \{0.001, 0.0007, 0.0005, 0.0003, 0.0001, 0.00005, 0.00001\}$, the exponential decay step for $\alpha$ is selected from $\{1000, 750, 500\}$ and the exponential decay rate for $\alpha$ is selected from $\{0.95, 0.9, 0.85, 0.8\}$. The actor $\pi_\theta$ and critic $Q_\nu$ in the GIL (i.e., Alg. 1) are trained using deep deterministic policy gradient (DDPG) Lillicrap et al. (2015) where the discounting factor $\gamma = 0.99$. We choose the behavioural policy $\beta$ in GIL as

$$\beta(\mathbf{s}) = \begin{cases} \pi_\theta(\mathbf{s}) & \text{with probability } p_1, \\ \text{missing indicator } \mathbf{m} & \text{with probability } p_2, \\ \text{random action} & \text{with probability } p_3. \end{cases} \tag{42}$$

The learning rates of the actor $\alpha_\pi$ and the critic $\alpha_Q$ are selected by performing grid search from $\{0.0005, 0.0001, 0.00005, 0.00001\}$ and $\{0.001, 0.0005, 0.0001\}$ respectively. To train the imputation-free models using GIL-H/GIL-D, we follow the same grid search for $\alpha$ along with its decay steps and decay rates. From the implementation perspective, we replace the missing entries, in the inputs $\mathbf{x}$, with a placeholder value before feeding them into the model. This can avoid value errors thrown by Tensorflow if the input vectors contain NaN's. However, note these values will not be used to update the parameters. For the state-of-the-art baselines – GAIN[5], MIWAE[6], GPVAE[7] and BRITS[8] – we use the implementations published on the Github by the authors. Adam optimizer is used to train all the prediction models for baselines, or to compute $(\partial E / \partial \mathbf{W})_{\text{SGD}}$ for GIL, GIL-H and GIL-D. All the models are trained using a batch size of 128. The details of selecting the other hyper-parameters of each case study can be found in the corresponding sub-section below.

## D.1    MIMIC-III

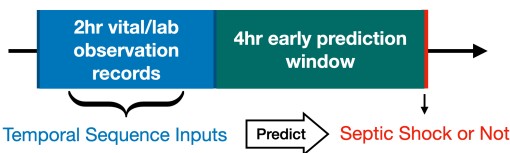

Figure 4: Graphical depiction of the 2-hour observation window and 4-hour early prediction window (EPW) considered in this case study.

**Dataset Formulation**    The MIMIC-III[9] contains EHRs obtained from roughly 40,000 patients who stayed in the ICUs in the Beth Israel Deaconess Medical Center between 2001 and 2012 (Johnson et al., 2016) and septic shock is a severe type of sepsis that results in above 40% mortality rate. Figure 4 illustrates the 2-hour observation window, 4-hour early prediction window (EPW) and the relation between inputs and targets for predictions. Note that each work related to septic shock predictions adopts a slightly different strategy in terms of selecting lab/vital attributes and the lengths of the observation and EPW (Mao et al., 2018; Darwiche & Mukherjee, 2018; Yee et al., 2019; Sheetrit et al., 2017; Liu et al., 2019; Khoshnevisan et al., 2020; Fleuren et al., 2020). In this work, the 4-hour EPW allows including sufficient number of subjects which can ensure statistical significance of the results, while the smaller observation window keeps the task challenging. To formulate the training and testing dataset, we first selected 14 commonly used attributes for predictions as suggested in previous works (Sheetrit et al., 2017; Fleuren et al., 2020; Khoshnevisan et al., 2020), which are constituted by vital signs including temperature, respiratory rate, heart rate, systolic blood pressure,

---

[5] https://github.com/jsyoon0823/GAIN

[6] https://github.com/pamattei/miwae

[7] https://github.com/ratschlab/GP-VAE

[8] https://github.com/caow13/BRITS

[9] Data acquired from https://mimic.mit.edu. Access to this dataset follows the MIT-CLP license.

mean arterial pressure, peripheral oxygen saturation (or SpO$_2$), fraction of inspired oxygen (or FIO$_2$), and lab tests including white blood cell count, serum lactate level, platelet count, creatinine, bilirubin, bandemia of white blood cells, blood urea nitrogen. To form the septic shock cohort, we first identify the patients who were diagnosed with sepsis by indexing with the International Classification of Diseases 9 (ICD-9) code 995.91 and 995.92. Then the patients with septic shock history are selected using ICD-9 code 785.52 with their septic shock onset time determined following the third international consensus for sepsis and septic shock (Sepsis-3) standard (Singer et al., 2016). Finally, we remove the patients whose septic shock onset time was less than 6 hours after admission to the ICUs since the data is not enough to be formulated as sequences pertaining to the 2-hour observation and 4-hour prediction window. Consequently, a total of 1,083 septic shock patients are identified. To form the non-shock (or control) cohort, first 1,083 patients are randomly sampled from all admissions who have at least 2 hours of records excluding the ones who are in the septic shock cohort, then a 2-hour time frame is randomly selected to be used as the observation window. All patients selected following the above procedure are split into 8:2 to formulate the training and testing datasets, which are referred to as the varied-length (Var-l.) sequences in Section 4.2. In the test set, the number of subjects associated with positive and negative labels, respectively, are selected to be equivalent.

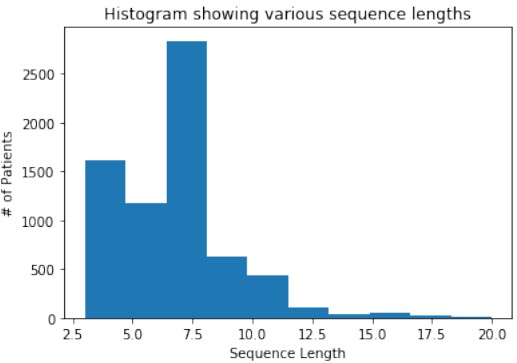

Figure 5: Histogram of the sequences lengths of the MIMIC-III dataset (with maximum length truncated at 20). It can be observed that most of the sequences have length $\leq 8$.

**Formulation of Fix-l. Sequences**   To formulate the fixed-length (Fix-l.) sequences, we follow this protocol – if sequence length is less than 8, we pad it with the latest recording available until the length reaches 8, otherwise we truncate the length to 8 by discarding the data from that point on-wards. We specifically choose this threshold because most of the sequences have length $\leq 8$, as shown in Figure 5.

**Type of Missingness**   The missing data in this dataset can be considered as a mixture of MCAR, MAR and MNAR. Specifically, recordings from human mistakes or malfunctioned equipment manifest as MCAR, the dependency across vital signs and lab results give rise to MAR (*e.g.*, specific lab tests are ordered only if abnormalities found in the related vital readings or other lab results), and the mismatch among the sampling frequency of some periodically recorded attributes such as temperature (obtained hourly) and blood test (conducted daily) can be considered as MNAR (Little & Rubin, 2019; Mamandipoor et al., 2019).

**Training Details**   To train the prediction models for both GIL and baselines, we use maximum training steps of 2,000 and 4,000 for varied-length and fixed-length sequences respectively. For imputation of the missing values, we train the imputation method MIWAE using the learning rates $\{0.001, 0.0001\}$ with other hyper-parameters provided by the authors in the code. To train GAIN, we use the default learning rates provided by the authors to train the generator and discriminator. GAIN contains one hyper-parameter that balances between the two losses $\mathcal{L}_G$ and $\mathcal{L}_M$ defined in Yoon et al. (2018) which is selected from $\{0.1, 1, 10, 100\}$. To train GP-VAE on the Fix-l. case, the set of hyper-parameters we use contain $latent\_dim = \{6, 12, 35\}, encoder\_size = 256, 256, decoder\_size = 256, 256, 256, window\_size = \{3, 4, 6\}, beta = \{0.2, 0.5, 0.8\}, sigma = 1.005, length\_scale =$

$\{2, 4, 7\}$. To train BRITS we used the recommended $impute\_weight = 0.3, label\_weight = 1.0$. The term $D(\zeta^+, \zeta^-)$ included in the reward function for GIL-D is defined as in the follows. Assume that in each training epoch $b/2$ inputs with label 0 and $b/2$ inputs with label 1 are sampled from the dataset, where $b$ is the batch size. We use $F^+ = (\zeta_1^+, \ldots, \zeta_{b/2}^+) \in \mathbb{R}^{b/2 \times e}$ to denote the features associated with positive labels (*i.e.*, 1) in the current batch and $F^- = (\zeta_1^-, \ldots, \zeta_{b/2}^-) \in \mathbb{R}^{b/2 \times e}$ are , where $e$ is the dimension of each individual feature $\zeta$. Then we define $D$ as

$$MSE(F^+[0:b/4], F^-[0:b/4]) + MSE(F^+[b/4:b/2], F^-[b/4:b/2])$$
$$-MSE(F^+[0:b/4], F^+[b/4:b/2]) - MSE(F^-[0:b/4], F^-[b/4:b/2]), \tag{43}$$

where the slicing index follows the syntax from Python, *e.g.*, $F^+[0:b/4]$ corresponds to the 0-th to $(b/4 - 1)$-th rows of $F^+$.

## D.2 OPHTHALMIC DATA

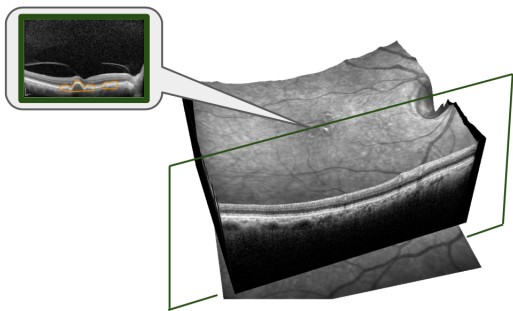

Figure 6: Example of a 3D OCT volume scan and a 2D OCT image slice from the volume scan.

**Dataset Introduction**  This dataset is originally constituted by OCT images, CFP images and patient EHRs including demographic information and test results related to diabetes. A total of 1,148 subjects are included in this dataset. OCT refers to the two- or three-dimensional images capturing the retinal architectures of the eyes that are scanned by low-coherence lights. The 3D OCT image is usually called the volume scan and it is constituted by 61 or 101 2D image slices depending on the spec of the scanner. Examples of 3D and 2D OCT scans are shown in Figure 6. In this experiment the volume scans we use contain the 61 slices, while we only consider the center slice (31th) 2D image. The CFP images are the color images captured by a fundus camera showing the condition of the interior surface of the eye, where an example is shown in Figure 7.

**Dataset Formulation**  To formulate the data into tabular inputs, the 2D OCT and CFP retinal images are first fed in to two inception-v3 (Szegedy et al., 2016) CNNs, pre-trained with similar images provided by (Kermany et al., 2018) and (Kaggle, 2015) respectively, and the image feature vectors with dimension of 2,048 each are output by the global average pooling layer. Then the patient EHR information include age (integer), sex (boolean), length of diabetic history (integer), A1C result (integer), which measures blood sugar levels, and if insulin has been used (boolean) constitute a 5-dimensional vector that is concatenated to the end of the OCT and CFP feature vectors. We split all the subjects into a training cohort and a testing cohort following a ratio of 9:1. In the test set, the number of subjects associated with positive and negative labels, respectively, are selected to be equivalent. The raw images from the training cohort are augmented through cropping and rotation before feeding into inception-v3. All the data from the testing cohort are not augmented. Figure 7 illustrates how the training and testing inputs are formulated, as well as the input-output relation of the prediction models.

**Training Details**  To train the prediction models for both GIL and baselines, we use maximum training steps of 2,000. For imputation of the missing values, MIWAE is trained using the learning rates $\{0.001, 0.0001\}$ with other hyper-parameters provided by the authors in the

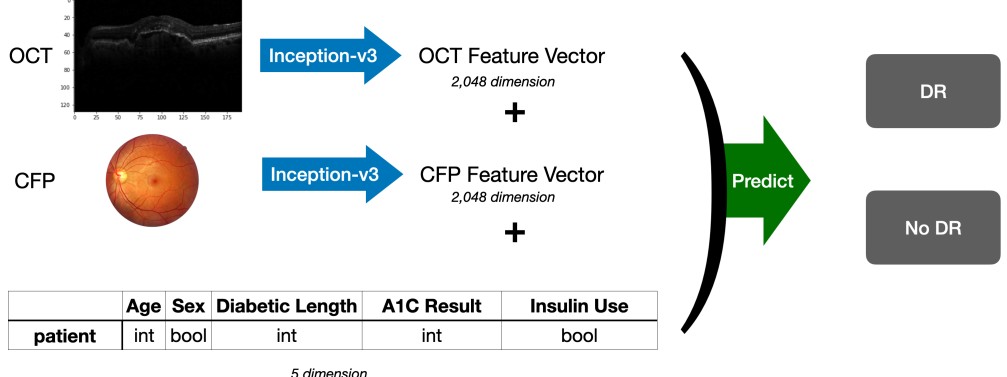

Figure 7: Diagram showing the pipline of this experiment.

code. GAIN is trained using hyper-parameters selected from $\{0.1, 1, 10, 100\}$. GP-VAE is trained by applying the arguments $latent\_dim = 256, encoder\_size = 256, 256, decoder\_size = 256, 256, 256, window\_size = 3, beta = 0.8, sigma = 1$ to its code published in Github (`https://github.com/ratschlab/GP-VAE`). MF, EM and kNN are not included in this experiment due to the high dimensionality of the inputs which makes these algorithms very computational inefficient as imputation results cannot be produced within days.

### D.3 MNIST

**Training Details**  To train the prediction models for both GIL and baselines, we use maximum training steps of 1,0000. For imputation of the missing values, MIWAE is trained using the learning rates $\{0.001, 0.0001\}$ with other hyper-parameters provided by the authors in the code. GAIN is trained using hyper-parameters selected from $\{0.1, 1, 10, 100\}$. GP-VAE is trained by applying the arguments $latent\_dim = 256, encoder\_size = 256, 256, decoder\_size = 256, 256, 256, window\_size = 3, beta = 0.8, sigma = 1$ to its code published in Github (`https://github.com/ratschlab/GP-VAE`). MF, EM and kNN are not included in this experiment due to the high dimensionality of the inputs which makes these algorithms very computational inefficient as imputation results cannot be produced within days.

**Additional Results**  We also trained GIL-D on the MCAR version of MNIST dataset and the performance are shown in Table 5 below.

Table 5: Accuracy for GIL-D on the MCAR version of MNIST dataset. Standard deviations are in subscripts ($\times 10^{-3}$).

| Missing Rate | 50% | 70% | 90% |
|---|---|---|---|
| Acc. | $96.29_{0.09}$ | $93.35_{0.9}$ | $78.47_{3.9}$ |

### D.4 PHYSIONET

Table 6: Accuracy, AUC and average precision (AP) obtained from the Physionet dataset

| | GIL | GIL-H | GAIN | GP-VAE | MIWAE | BRITS | Mean | Zero |
|---|---|---|---|---|---|---|---|---|
| Acc. | 87.45 | 87.38 | 86.23 | 86.85 | 87.12 | 86.20 | 86.87 | 87.15 |
| AUC | 82.20 | 82.00 | 78.02 | 82.21 | 83.47 | 81.30 | 82.57 | 81.14 |
| AP | 49.19 | 48.64 | 39.89 | 47.05 | 49.31 | 43.57 | 48.30 | 47.37 |

Other than MIMIC-III, we also tested on a smaller scaled ICU time-series from 2012 Physionet challenge Silva et al. (2012) which contain data obtained from 12,000 patients. We use the data pre-processed and open-sourced by Fortuin et al. (2020).[10] For each patient the values of 35 different attributes (*e.g.,* blood pressure, temperature) are recorded over a 48-hour window. As a result, the data for each patient can be formulated into a matrix in $\mathbb{R}^{48 \times 35}$, *i.e.*, the sequence length across patients are the same. This dataset has an overall 78.5% missing rate and a binary label is assigned to each patient where 87% of them are 0's and 13% are 1's. Therefore we include average precision (AP) which is calculated from the precision-recall curve as an additional metric to evaluate the performance toward imbalanced labels. For this dataset, we consider a 1024-unit LSTM layer for encoding and a dense output layer for inference. It can be observed from Table 6 that although our method achieves the highest accuracy and 2-nd highest AP, while most methods perform very close to mean- and zero-imputation. This could be caused by the its simple structure, as all the sequences share the same time horizon, which significantly reduces the difficulties for the classification task. Moreover, the highly imbalanced labels and the small population result in the performance to be hardly distinguishable across different methods. And this is the reason that we focus on the MIMIC-III dataset to study the strengths and shortcomings of the methods.

---

[10]https://github.com/ratschlab/GP-VAE

# E PROOF FOR PROPOSITION 1

Here we prove a generalized version of Proposition 1 as stated in the following proposition, which shows that the gradient of the loss function $E$ w.r.t. any layer in the MLP can be written in the outer product format.

**Proposition 3** *Given an MLP as in (1) and a smooth loss function $E(\hat{\mathbf{y}}|\mathbf{y})$, the gradients for any hidden layer $\mathbf{W}_i$, $i \in [1, k]$ can be represented as*

$$\frac{\partial E}{\partial \mathbf{W}_i} = \boldsymbol{\Delta}_i \mathbf{q}_{i-1}^\top, \tag{44}$$

*where $\mathbf{q}_{i-1} = \phi_{i-1}(\mathbf{W}_{i-1}\mathbf{q}_{i-2})$ is the output from the $i-1$-th layer with $\mathbf{q}_0 = \mathbf{x}$, and $\boldsymbol{\Delta}_i = \mathbf{W}_{i+1}^\top \boldsymbol{\Delta}_{i+1} \odot \phi_i'(\mathbf{W}_i \mathbf{q}_{i-1})$ with $\boldsymbol{\Delta}_{out} = \frac{\partial E}{\partial \hat{\mathbf{y}}} \odot \phi_{out}'(\mathbf{W}_{out}\mathbf{q}_{2k})$ and $\odot$ denotes the element-wise (Hadamard) product.*

Now we prove Proposition 3.

We consider the MLP characterized by

$$\hat{\mathbf{y}} = \phi_{out}(\mathbf{W}_{out}\phi_k(\mathbf{W}_k \ldots \phi_2(\mathbf{W}_2\phi_1(\mathbf{W}_1\mathbf{x})))) \tag{45}$$

where $\mathbf{x} \in \mathbb{R}^d$ represents the input to the model, $\mathbf{W}_i$ is the weight matrix for the $i$-th hidden layer, and $\phi_i$ is the activation function of the $i$-th hidden layer.

We start with deriving the derivatives for the output layer

$$\frac{\partial E}{\partial \mathbf{W}_{out}} = \frac{\partial E}{\partial \hat{\mathbf{y}}} \frac{\partial \hat{\mathbf{y}}}{\partial \mathbf{W}_{out}} \tag{46}$$

$$= [\frac{\partial E}{\partial \hat{\mathbf{y}}} \odot \phi_{out}'(\mathbf{W}_{out}\mathbf{q}_k)] \cdot \mathbf{q}_k^\top \tag{47}$$

$$= \boldsymbol{\Delta}_{out} \cdot \mathbf{q}_k^\top, \tag{48}$$

where $\mathbf{q}_k$ is the output from the $k$-th hidden layer.

Now we show the derivative of $E$ w.r.t. the $k$-th hidden layer

$$\frac{\partial E}{\partial \mathbf{W}_k} = \frac{\partial E}{\partial \hat{\mathbf{y}}} \frac{\partial \hat{\mathbf{y}}}{\partial \mathbf{q}_k} \frac{\partial \mathbf{q}_k}{\partial \mathbf{W}_k} \tag{49}$$

$$= \mathbf{W}_{out}^\top [\frac{\partial E}{\partial \hat{\mathbf{y}}} \odot \phi_{out}'(\mathbf{W}_{out}\mathbf{q}_k)] \cdot \frac{\partial \mathbf{q}_k}{\partial \mathbf{W}_k} \tag{50}$$

$$= \mathbf{W}_{out}^\top \boldsymbol{\Delta}_{out} \cdot \frac{\partial \mathbf{q}_k}{\partial \mathbf{W}_k} \tag{51}$$

$$= [(\mathbf{W}_{out}^\top \boldsymbol{\Delta}_{out}) \odot \phi_k'(\mathbf{W}_k \mathbf{q}_{k-1})] \cdot \mathbf{q}_{k-1}^\top \tag{52}$$

$$= \boldsymbol{\Delta}_k \cdot \mathbf{q}_{k-1}^\top \tag{53}$$

We now prove Proposition 3 by induction. First assume that for $j \in [2, k]$, the derivative of $E$ w.r.t. the $j$-th hidden layer is

$$\frac{\partial E}{\partial \mathbf{W}_j} = [(\mathbf{W}_{j+1}^\top \boldsymbol{\Delta}_{j+1}) \odot \phi_j'(\mathbf{W}_j \mathbf{q}_{j-1})] \cdot \mathbf{q}_{j-1}^\top \tag{54}$$

$$= \boldsymbol{\Delta}_j \cdot \mathbf{q}_{j-1}^\top. \tag{55}$$

Moreover, let us assume that

$$\frac{\partial E}{\partial \mathbf{q}_j} = \mathbf{W}_{j+1}^\top [\frac{\partial E}{\partial \mathbf{q}_{j+1}} \odot \phi_{j+1}'(\mathbf{W}_{j+1}\mathbf{q}_j)] \tag{56}$$

$$= \mathbf{W}_{j+1}^\top \boldsymbol{\Delta}_{j+1}. \tag{57}$$

Now we need to show that for $j - 1$, it holds that

$$\frac{\partial E}{\partial \mathbf{W}_{j-1}} = [(\mathbf{W}_j^\top \mathbf{\Delta}_j) \odot \phi'_{j-1}(\mathbf{W}_{j-1}\mathbf{q}_{j-2})] \cdot \mathbf{q}_{j-2}^\top, \tag{58}$$

$$\frac{\partial E}{\partial \mathbf{q}_{j-1}} = \mathbf{W}_j^\top [\frac{\partial E}{\partial \mathbf{q}_j} \odot \phi'_j(\mathbf{W}_j\mathbf{q}_{j-1})] \tag{59}$$

$$= \mathbf{W}_j^\top \mathbf{\Delta}_j. \tag{60}$$

We first prove that (60) holds, i.e.,

$$\frac{\partial E}{\partial \mathbf{q}_{j-1}} = \frac{\partial E}{\partial \mathbf{q}_j} \frac{\partial \mathbf{q}_j}{\partial \mathbf{q}_{j-1}} \tag{61}$$

$$= \mathbf{W}_{j+1}^\top \mathbf{\Delta}_{j+1} \frac{\partial \mathbf{q}_j}{\partial \mathbf{q}_{j-1}} \tag{62}$$

$$= \mathbf{W}_j^\top [\mathbf{W}_{j+1}^\top \Delta_{j+1} \odot \phi'_j(\mathbf{W}_j\mathbf{q}_{j-1})] \tag{63}$$

$$= \mathbf{W}_j^\top \mathbf{\Delta}_j, \tag{64}$$

which is equivalent to (60).

To prove that (58) holds, we start with

$$\frac{\partial E}{\partial \mathbf{W}_{j-1}} = \frac{\partial E}{\partial \mathbf{q}_j} \frac{\partial \mathbf{q}_j}{\partial \mathbf{q}_{j-1}} \frac{\partial \mathbf{q}_{j-1}}{\partial \mathbf{W}_{j-1}} \tag{65}$$

$$= \mathbf{W}_{j+1}^\top \mathbf{\Delta}_{j+1} \frac{\partial \mathbf{q}_j}{\partial \mathbf{q}_{j-1}} \frac{\partial \mathbf{q}_{j-1}}{\partial \mathbf{W}_{j-1}} \tag{66}$$

$$= \mathbf{W}_j^\top [\mathbf{W}_{j+1}^\top \Delta_{j+1} \odot \phi'_j(\mathbf{W}_j\mathbf{q}_{j-1})] \frac{\partial \mathbf{q}_{j-1}}{\partial \mathbf{W}_{j-1}} \tag{67}$$

$$= \mathbf{W}_j^\top \mathbf{\Delta}_j \frac{\partial \mathbf{q}_{j-1}}{\partial \mathbf{W}_{j-1}} \tag{68}$$

$$= [(\mathbf{W}_j^\top \mathbf{\Delta}_j) \odot \phi'_{j-1}(\mathbf{W}_{j-1}\mathbf{q}_{j-2})] \cdot \mathbf{q}_{j-2}^\top, \tag{69}$$

which is equivalent to (58).

## F  ADDITIONAL RELATED WORKS

**RL for Optimization.**    This work is also related to approaches that use RL to solve optimization problems, or to improve existing solvers. For example, Boyan & Moore (2000) proposes a framework to improve the efficiency of local search algorithms by predicting the search outcomes and biasing toward directions that provide higher returns, which is approached by estimating value functions of the MDP corresponding to the optimization problem. Recently, Li & Malik (2016) trains RL agents to perform gradient descent steps over linear regression and neural network models. Similarly, Wei et al. (2020) introduces a policy network to automatically determine parameters for the plug-and-play frameworks (Sreehari et al., 2017) which can solve inverse imaging problems. Note that these methods require complete inputs and cannot be adapted to the problem we consider.

## G  ADDITIONAL ABLATION STUDY

In this section, we compare GIL against an ablation baseline that applies the importance directly to the inputs $\mathbf{x}$, instead of the gradient space. Specifically, suppose a differential parametric function $h_\psi(\mathbf{x})$, parameterized by $\psi$, is multiplied with $\mathbf{x}$ element-wise. Then, as opposed to (4), the gradient updates w.r.t. the encoding layers can formulated as

$$\mathbf{W}'_{enc} \leftarrow \mathbf{W}_{enc} - \alpha \mathbf{\Delta} \cdot (\mathbf{x}^\top \odot h_\psi(\mathbf{x})^\top). \tag{70}$$

Consequently, all model elements (*i.e.*, $h_\psi$, $\mathbf{W}_{enc}$ and $\mathbf{W}_{inf}$) could be trained by gradient descent, without introducing RL policies.

Table 7: Performance of the ablation model over the ophthalmic and MNIST datasets

|  | Ophthalmic | | MNIST | |
| --- | --- | --- | --- | --- |
|  | 25% M.R. | 35% M.R. | 70% M.R. | 90% M.R. |
| Acc. | $84.50_{1.49}$ | $80.99_{1.09}$ | $93.22_{0.6e-03}$ | $78.3_{1.8e-03}$ |
| AUC | $89.78_{1.68}$ | $87.26_{2.53}$ | - | - |

In Table 7 it shows the performance of the model described above over the ophthalmic dataset, as well as MNIST digits with 70% and 90% missing rate (M.R.). Specifically, $h_\psi(\mathbf{x})$ is captured by a neural network with the same architecture as the policy $\pi_\theta$ in GIL. The training and testing are conducted following the same convention (*e.g.*, random seeds that determine the missing entries and hyper-parameter search) as introduced in Sections 4.3, 4.4 and Appendix D.2, D.3. It can be observed that the ablation baseline attained similar performance as to GIL-H and zero imputation. The reason that leads to these results would be that the gradients for training $h_\psi(\mathbf{x})$ still follow the outer product format $\mathbf{\Delta} \cdot \mathbf{x}^\top$ which are left to be accounted for. Specifically, our method is built on top of the idea, and the experiments in Section 4 also support that if part of the inputs are missing they may not provide sufficient information to train the prediction models, given the outer product format of the gradients. Similarly, the gradients for $h_\psi(\mathbf{x})$ also follow this format and $h_\psi(\mathbf{x})$ may not be properly trained directly on incomplete inputs $\mathbf{x}$, which could in general limit the overall performance of the prediction model.

**Additional rationales for using RL to obtain gradient importance in GIL.**    The ablation baseline above is closely related to visual attention (VA) models as discussed in Appendix C, and could be seen as a preliminary version of them where $h_\psi(\mathbf{x})$ is used to re-weight elements in the inputs $\mathbf{x}$. VA techniques benefit from the fact that features learned by CNNs are spatially correlated to the inputs, so the attentions could be directly applied to learned features. To achieve this, VA formulates the objective as an ELBO which is shown equivalent to the objective of a basic policy gradient RL algorithm, REINFORCE (Mnih et al., 2014). However, it was not clear how such VA methods could be adapted to the problem we consider, as the features learned by MLPs or LSTMs are not spatially correlated with inputs.

**More on importance versus attentions.**    The other baseline, BRITS (Cao et al., 2018), uses bidirectional RNN to process time-series, with attentions applied to the hidden states of RNNs, followed by optimizing over imputation and prediction objectives jointly during training. BRITS is considered as the state-of-the-art method that could predict with incomplete time-series inputs *end-to-end*. However, we showed that BRITS is outperformed by our method on the MIMIC dataset; see Table 1. The reason would be that BRITS requires masking off part of the observed inputs to constitute the imputation objective; thus, the information provided for model training is even more limited given the intrinsic $> 70\%$ M.R. of MIMIC.

