# OpenReview forum: "Gradient Importance Learning for Incomplete Observations"
_ICLR.cc/2022/Conference — ICLR 2022 Poster_

### Official Review · Reviewer_3AZM · 2021-10-31

**Correctness:** 4
**Technical Novelty And Significance:** 3
**Empirical Novelty And Significance:** 3
**Recommendation:** 6
**Confidence:** 4

**Main Review:**

I have several concerns as follows:

1. The motivation of the paper is not so clear. Specifically, I'm not fully convinced by the arguments between Eq. (2) and Eq. (3). I understand how the missing values can affect the learning of the parameters in the encoder according to Eq. (2). However, adding an element-wise multiplicative parameter matrix A seems quite arbitrary.

1.1 Why choose this specific form?

1.2 Note that each sample may have a different mask sampled randomly. Can a shared matrix across all data samples solve the problems caused by the missing values with diverse patterns? This is counter-intuitive.

1.3. Why it must be on the gradient level, not the parameter level? If the matrix A is prefixed, using the gradient in Eq. (3) is equivalent to reparameterizing the encoder layer by multiplicating the same A, right? Then why not directly optimize all parameters using SGD without adding an RL procedure. I know the final parameters are different by using different optimization strategies. But why this gradient level modification with RL is preferable for this task?

2. I note the authors discussed the limitation of the proposed models, mainly about CNN. I'm also wondering the same questions about other popular techniques used in deep learning, including but not limited to batch normalization and residual structures.

3. How does the additional RL training procedure affects the efficiency of the training? It is necessary to compare the time and memory complexities and training stabilities.

4. What is learned in matrix A? Intuitively, different mask distributions will affect matrix A but how? Any intuition behind this?

5. For the experiments, I didn't see how the hyperparameters including the model architectures are selected. For instance, does the MLP structures used in this paper exactly the same as the ones used in SOTA methods? How do the additional hyperparameters set?

6. A very basic "imputation-free" baseline is to directly train a model to predict the label given an incomplete input. For instance, on the MNIST dataset, I wonder about the results of widely used CNN models  (e.g. Alexnet, Resnet-26) on the same data. I notice the limitation of the paper on CNNs while it is not the reason to ignore such a natural baseline.


**Summary Of The Paper:**

The paper proposes gradient importance learning for predicting labels on incomplete data. The problem is: during both training and testing, certain dimensions of the samples are masked randomly and the masks (i.e. the "missing indicators") are known for the algorithm. Among existing methods in this problem, the paper claims that it is the first "imputation-free" way to solve this problem. The paper focuses on MLP and LSTM and adds an (element-wise) multiplicative parameter matrix on the gradient of the first layer (the one that is mostly closed to the data). The matrix is trained using reinforcement learning. The authors also connect its proposal to visual attention. Experiments are conducted on tabular, time series, and a simple image dataset.


**Summary Of The Review:**

I raise several questions about the motivation, clarity, and experiments of the paper. I believe the current version should be significantly improved to reach the acceptance of ICLR.

**post discussion with the authors**

I thank the authors for this productive reviewing process. I increased my score because my major concern has been addressed.

---

> ### Author Response · Authors · 2021-11-16
> **Thank you for the comments. Please see our responses below. [Part 2]**
>
>
> **Response to 6**
>
> Thank you for this comment. In the problem definition we stated that this works considers missingness problems in tabular and time-series data, where MLPs and LSTMs would be sufficient for such tasks. The reason we chose to use MNIST digits as a test base is due to its simple structure and substantial amount where the strengths of our method can be illustrated straightforwardly without domain expertise, as opposed to the other two medical datasets. Specifically, in Figure 1 it shows that GIL can learn more expressive feature representations than baselines. Also in Table 4 it helps illustrate the negative correlation between imputation performance and prediction performance with statistical significance. We also adopted the zero-imputed MNIST digits as inputs to MLPs models which serves as the vanilla baseline, similar to what was suggested by the reviewer. If CNNs were to be included into such a comparison it may achieve superior performance since it can capture the spatial correlations from MNIST digits, which could not be learned by MLPs and LSTMs. However, CNNs are not commonly used toward the tabular data or time-series as considered in missing data problems.

---

> > ### Comment · Reviewer_3AZM · 2021-11-18
> > **Thanks. More questions.**
> >
> > Thank you for the detailed feedback, especially for clarifying my misunderstanding on how $A$ works.
> >
> > I have some further questions about the necessity of the form of $A$ and the usage of RL. Now assume we introduce a differentiable parametric model $f_\theta(x)$ (in analogy to the policy $\pi_\theta$) that outputs an individual mask $A$ for each sample $x$. Instead of multiplying it at the gradient level, we can directly multiply it at the parameter level. Then, the gradient w.r.t. the parameters in the encoders are the same as Eq. (3). However, in this formulation, the parameters in $f_\theta$, i.e. $\theta$ can be directly optimized using gradient descent together with the encoder. In comparison, why GIL is preferable to this way, which seems natural, efficient, and easy to implement and understand?
> >
> > I still worry about the necessity of the framework and the efficiency issue caused by the potential overcompleteness. I'm happy to increase my score if the concerns can be addressed.

---

> > > ### Author Response · Authors · 2021-11-19
> > > **Additional experiments showing our method outperforms the model suggested by reviewer. Followed by discussion.**
> > >
> > > Thank you for the thoughtful comment. We have run additional experiments which show that our approach outperforms such a model. In what follows, some discussions are made in regards to addressing the results and reviewer’s concern.
> > >
> > > **Experimental Details and Results**
> > >
> > > We parameterize $f_\theta(x)$ as a neural network following the same architecture as the policy network used in our approach. The output of $f_\theta(x)$ is multiplied element-wise with the input data $x$ before feeding into the prediction model, such that RHS of equation (4) becomes $W_{enc} - \alpha \Delta \cdot (x^T \odot f_\theta(x)^T)$, as suggested by the reviewer. We tested this model on the ophthalmic dataset as well as MNIST digits with 70% and 90% missing, following the same convention as training GIL methods (e.g., same random seeds for generating masked data and hyper parameter set for training prediction models). The results shown in table below show that it’s performance is close to zero imputation or the ablation baseline GIL-H.
> > >
> > > |     | 25% Missing Ophthalmic | 35% Missing Ophthalmic | 70% Missing MNIST  | 90% Missing MNIST |
> > > |-----|------------------------|------------------------|--------------------|-------------------|
> > > | ACC | 84.50 +- 1.49 %        | 80.99 +- 1.09 %        | 93.22 +- 0.6E-03 % | 78.3 +- 1.8E-03 % |
> > > | AUC | 89.78 +- 1.68 %        | 87.26 +- 2.53 %        | -                  | -                 |
> > >
> > > **Discussion**
> > >
> > > 1. The reason that leads to these results would be that the gradients for training $f_\theta(x)$ still follow the outer product format $\Delta \cdot x^T$, and are left to be accounted for, similar to the zero imputation baseline or GIL-H. Specifically, our method is built on top of the idea, and the experiments in the paper also support that if part of the inputs are missing they may not provide sufficient information to train the prediction models, given the outer product format of the gradients. Similarly, the gradients for $f_\theta(x)$ also follow this format and $f_\theta(x)$ may not be properly trained directly on incomplete inputs $x$, which could in general limit the performance of the prediction model overall.
> > >
> > > 2. The model suggested by the reviewer is closely related to visual attention models as discussed in Appendix C, and could be seen as a preliminary version of them where $f_\theta(x)$ is used to re-weight elements in the inputs $x$. Visual attention techniques benefit from the fact that features learned by CNNs (after global average pooling layer) are spatially correlated to the inputs, so the attentions could be directly applied to learned features. **To achieve this they formulate the objective as an ELBO, which is shown equivalent to the objective of REINFORCE (a very basic policy gradient method in RL) [1].** However, it was not clear how such visual attention methods can be adapted to the problem we consider, as in MLPs/LSTMs the learned features are not spatially correlated with inputs.
> > >
> > > 3. One of the baselines, BRITS [2], uses bidirectional RNN to process time-series, with attentions applied to the hidden states of RNNs, followed by optimizing over imputation and prediction objectives jointly during training. BRITS is considered as the state-of-the-art method that could predict with incomplete time-series inputs in an *end-to-end* manner. However, we showed that BRITS is outperformed by our method on the MIMIC dataset; see Table 1. The reason would be that BRITS requires masking off part of the observed inputs to constitute the imputation objective; thus, the information provided for model training is even more limited given the intrinsic ~70% missing rate of MIMIC.
> > >
> > > We truly appreciate your efforts reviewing and commenting on our paper. We hope this response addresses your concerns well and we are happy to respond to any other further comments!
> > >
> > > **References**
> > >
> > > [1] Mnih, V., Heess, N. and Graves, A., 2014. Recurrent models of visual attention. In Advances in neural information processing systems (pp. 2204-2212).
> > >
> > > [2] Cao, W., Wang, D., Li, J., Zhou, H., Li, L. and Li, Y., 2018. BRITS: Bidirectional Recurrent Imputation for Time Series. Advances in Neural Information Processing Systems, 31, pp.6775-6785.

---

> > > > ### Comment · Reviewer_3AZM · 2021-11-20
> > > > **Thanks**
> > > >
> > > > I have changed my score because the authors clarify my concerns. I hope the authors could incorparate the above results and discussion somewhere in the next version. I thank the authors for this productive reviewing process.

---

> > > > > ### Author Response · Authors · 2021-11-22
> > > > > **We have now included the results and discussion above into the latest revision.**
> > > > >
> > > > > We greatly appreciate the timely responses from the reviewer, as well as the efforts reading through our replies and following up with insightful comments. As suggested by the reviewer, the results and discussions we made in the response earlier are now included in to Appendix G of the paper.

---

> ### Author Response · Authors · 2021-11-16
> **Thank you for the comments. Please see our responses below. [Part 1]**
>
> Thank you for providing detailed feedback and we truly appreciate your efforts for reviewing. We found that some details in regards to our methodology summarized by the reviewer (in Questions 1, 5, 6) are somewhat inconsistent with what has been presented in the manuscript, so we clarified them in the responses below (along with responses for other questions). If any part of the writing causes such misunderstandings, we would be happy to revise the paper such that our ideas can be introduced crystal clear. We also welcome any additional comments and will definitely try our best to address them in time.
>
> **Response to 1.1**
>
> Because the gradients for the encoding layer can be seen as the outer product between the inputs and the gradients propagated from deep layers. If part of the inputs are missing they may not provide useful information to train the prediction models, especially when the missing rate is high. By introducing the importance matrix $A$ it can augment the representations learned by the encoding layers, as shown in Figure 3.
>
> **Response to 1.2**
>
> The elements of the importance matrix $A$ are determined **on-the-fly, and are specific to the inputs** to the MLP/LSTM prediction models in the current training iteration, by following the RL policy (see Section 2.3 definition of action space); thus they are not the same across all the data samples.
>
> **Response to 1.3**
>
> As discussed in the previous question, the matrix $A$ is **not prefixed**, so our method is different from simply multiplying a matrix to the encoding layer. Moreover, another advantage of tackling this problem during training (in gradient space) would be -- once training is done and the prediction model needs to be evaluated, it can map inputs to predictions end-to-end (like any other neural networks would do), where the RL policies and the importance matrix $A$ are not involved since they are only used in the training stage.
>
> **Response to 2**
>
> Excellent point! The importance matrix $A$ only applies to the gradients of the encoding layer, so it would not affect batch normalization layers. In all the experiments, batch normalizations are applied to both the encoding layer, as well as all the inference layers, for the models trained by GIL and its variants (same for the prediction models for baselines). Residual structures are more commonly used to prevent gradients from vanishing in deep CNNs for computer vision tasks. However, for tabular/time-series MLPs and LSTMs would be sufficient and these models are less likely to experience such issues.
>
> **Response to 3**
>
> Thank you for the suggestion. We have evaluated our runtime/complexity against state-of-the-art baselines and please see our response to question 2 from reviewer azSY. The stabilities are evaluated in the manuscript where, for all the datasets, the standard deviations of the performance over different random seeds are reported in Table 1,2 and 3.
>
> **Response to 4**
>
> Thank you for this thoughtful question. The intuition of applying matrix $A$ to the gradient space is that sometimes the missing values may not need to be imputed but where and when the values are missing could be informative enough. As a result, our method approaches the missingness by using reinforcement learning to explore the best possible ways to guide the model toward capturing such 'missing patterns' and exploiting as much information as possible from the incomplete inputs to facilitate accurate predictions. The values in the importance matrix $A$ could be seen as measurements regarding how 'meaningful' each input entry would be to train a model to perform reasonable inference. If it places higher weights toward the entries that are not missing, then it could mean that those attributes are helpful for training the model. On the other hand, if the $A$ pays higher attention to the input entries that are missing, then it implies the missingness itself may already carry some useful information for prediction.
>
> **Response to 5**
>
> The prediction network for both GIL methods and all baselines share the same architecture and hyper-parameters, which is mentioned in Section 4.1 -- i.e.,
>
> *The imputed data from these baselines are fed into the prediction models that share the same architecture as the ones trained by GIL, for fair comparison.*
>
> Most baselines require imputing the data before prediction, and they are based on different ideas (e.g., VAEs, GANs etc.). So we follow the recommended architecture as proposed in the original works, and the hyper-parameter set for them is documented in Appendix D.

---

### Official Review · Reviewer_4Cb5 · 2021-11-01

**Correctness:** 3
**Technical Novelty And Significance:** 4
**Empirical Novelty And Significance:** 3
**Recommendation:** 6
**Confidence:** 3

**Details Of Ethics Concerns:**

I don't have any ethics concerns about this paper.

**Main Review:**

This paper targeted at an important problem in machine learning and proposed a novel idea to address it. The paper is well structured and easy to follow. The experimental results on the MIMIC-III datasets are encouraging.
My main concerns about this work are as follows:
1.	Imputation-based methods will not work well on data with informative missing, as in MIMIC-III, because patients in different conditions or different metrics are measured in different time scales. Thus, the missing patterns may contain some hidden information about the patients, and imputation may lead to a lost of the information. This can also be observed in Table 1, where most of the compared methods are worse than GIL-H (a heuristic that simply discards the gradients produced by the subset of input entries that are missing). Thus, there might be concerns that the proposed method will work well in time series data with informative missing but may not work equally well in other missing patterns, e.g., only random missing. For example, the results on the MNIST dataset seems to confirm that the proposed method cannot significantly outperform the baselines even with simple zero imputation.
2.	There might be an efficiency issue when using RL to weigh the importance of gradients. The training of the RL policies is generally time-consuming, so it will be also important to analyze the efficiency of the proposed method.

**Summary Of The Paper:**

This paper targeted at the missing data issue in time series data and proposed a imputation-free method to handle missing data. More specifically, the authors proposed a gradient importance learning method named GIL to weigh the gradients for different parameters using reinforcement learning. Experiments on one tabular dataset and two image datasets demonstrated the effectiveness of the proposed method.

**Summary Of The Review:**

Handling missing data is important in real machine learning problems and it is good to see that imputation-free method can also work well under this issue. Since there are some unanswered questions in the paper, I would like to give a slightly positive rating.

---

> ### Author Response · Authors · 2021-11-16
> **Thank you for the comments. Please see our responses below.**
>
> We thank you for the efforts and time spent on providing feedback to our submission. Please see below the responses to the questions posted in the review. As always, we are more than happy to answer any follow-up questions you may have!
>
> **Response to 1**
>
> Thank you for the thoughtful comment. Please note that the standard deviation reported for MNIST results (Table 3) are at scale of $\times 10^{-3}$, so the average performance of our method is still outside of the 99% confidence interval for all baselines in the MCAR case. Given how MNIST digits are formulated, zero imputation is expected to be a good heuristic -- as shown in Figure 1, though there are 70% of the pixels missing it is still possible to identify the outline of the digits roughly. Moreover, missing data is a practical problem encountered while analyzing data collected from real-world facilities, which tend to be affected by a mixture of different missingness mechanisms. For example, as discussed in Section 4.2, 4.3 and Appendix D.1, the MIMIC dataset contains missing data due to all three missingness mechanisms (i.e., MCAR, MAR and MNAR). The ophthalmic dataset also contains missingness due to MCAR and MAR.
>
> **Response to 2**
>
> Thank you for the suggestion. We have evaluated our runtime against state-of-the-art baselines and please see our response to question 2 from reviewer azSY.

---

> > ### Comment · Reviewer_4Cb5 · 2021-11-18
> > **Missingness mechanism**
> >
> > I think missingness mechanisms are key to the performance of imputation methods. Thus, I believe the superior performance of the proposed method may be related to the missingness pattern. For instance, there are larger improvements on the medical datasets, which I think could be due to the informative missing in these datasets, i.e., most missing data is MNAR. Similarly, the smaller improvements on the MNIST datasets may indicate that the advantage of the proposed method is not so substantial on MAR setting. Nevertheless, I believe the work is still interesting in terms of proposing a better solution for MNAR data. Therefore, I would like to keep my positive rating to this paper.

---

### Official Review · Reviewer_azSY · 2021-11-02

**Correctness:** 4
**Technical Novelty And Significance:** 3
**Empirical Novelty And Significance:** 4
**Recommendation:** 6
**Confidence:** 2

**Main Review:**

Pros
- Proposes novel method
- Outperforms baselines

Questions
- You say you use batches of size 128, but the description of the algorithm describes a scenario of batch size 1 if I understand correctly. How do you handle large batch sizes?
    - There also seems to be a slight mistake in the algorithm with the index i in $x_i$
- What are the compute requirements compared to the baselines?

Cons
- Complicated method, compared to mean imputation for example.
The method requires two different loops and naturally one wonders whether there wouldn't be a method that does only require one objective.
- I think the explanation of the method is a little lengthy and overly complicated, as one might assume knowledge of standard back propagation in MLPs.


**Summary Of The Paper:**

The paper proposes a method to train on data with missing features. It does so with a single model that can handle missing features, not with extra imputation methods. This is done by weighting the gradient update of the first weight matrix of the neural network with a vector a that is produced by an RL agent. This agent's reward in turn is the performance of the model after the update.

**Summary Of The Review:**

While I believe the objective of the paper is worth research: a neural network that works well with data missingness out of the box, I believe the method is a little complicated to be a practical tool and for other research to build upon it. Additionally, I want to note that, while I understand the algorithm well, I never worked with data missingness before.

---

> ### Author Response · Authors · 2021-11-16
> **Thank you for the comments. Please see our responses below. [Part 2]**
>
> **Response to Con1**
>
> Thank you for the thoughtful comment. We agree that mean/zero imputation would be a more straightforward way to deal with missing data, however, it has been shown in the case studies that **our method significantly outperforms such methods**. Moreover, please note that most baselines follow the two-step imputation-prediction framework, where an imputation model needs to be trained first followed by training the prediction model taking as input the imputed data; thus, these methods are not optimizing over a single objective. In contrast, our method can train the prediction model **end-to-end** which only involves a single training procedure -- this can be observed from Algorithm 1 which only loops over the training samples in the dataset. The baseline BRITS optimizes over the imputation loss and prediction loss jointly for recurrent neural networks toward time-series classification. Their performance is significantly outperformed by our method as shown in Table 1. Moreover, it is not considered 'imputation-free', since BRITS requires masking off part of the observed inputs to be used as training targets for their imputation objective.
>
> **Response to Con2**
>
> We could definitely simplify the part corresponding to MLPs in the paper. However, it may not be that straightforward for LSTMs (see Appendix A). So in the main text we use proposition 1 and equations (2), (3) to illustrate the design our method in the context of MLPs since it could illustrate our intuition and ideas behind it straightforwardly, such that readers could quickly capture the ideas while going through this section. The proposition and equations above also facilitate introducing important details in our approach in the section that follows. Details for LSTM models are relatively more complicated, so we put them in Appendix A.
>
> **Response to the Summary Of The Review**
>
> Thank you for all your comments. We list a few future works below, which could potentially leverage the framework we proposed.
>
> 1. Extend our method into inference tasks using image/video inputs with missingness. Missingness in images/videos usually leads to applications such as image super-resolution and compressive sensing, which are slightly different from how missingness are defined in tabular/time-series tasks. However, the common challenge for such tasks is that inputs may not contain full details (e.g., a compressed image) for the inference model to learn. Our method has shown that RL is capable of adjusting the gradients that are used to train the model toward improving the inference performance with imperfect inputs, and could possibly be extended to research topics in the image/video processing domain.
>
> 2. As mentioned in our response to Question 2 above, parallelizable RL frameworks could be leveraged in the future to improve the efficiency of our method. Moreover, if designed properly such frameworks can also facilitate capturing the uncertainty under missingness. For example, in the MIMIC-III dataset the ground-truths of all missing entries are not obtainable, which introduces uncertainty to the downstream inference tasks. A hypothesis would be that if such uncertainty could be captured during training, and being used to further calibrate the gradient estimates for updating the prediction models, they may lead the model toward learning more expressive feature representations and thus improved performance.

---

> ### Author Response · Authors · 2021-11-16
> **Thank you for the comments. Please see our responses below. [Part 1]**
>
> Thank you for recognizing our contributions and experimental results that support them. We also truly appreciate your effort in reviewing our submission. We address the two questions you asked in this reply (Part 1) and the other comments in Part 2. Please do not hesitate to follow up with any further questions/comments you might have and would be very happy to clarify them!
>
> **Response to Question1**
>
> Thank you for pointing out the typo in line 9 of the pseudo-code. We corrected it in the revision.
>
> To answer your question for batch-training -- similar to the common batch training techniques, RL policy $\pi_\theta$ can be batch-trained using the historically collected (s,a,r,s') tuples. Specifically, a replay buffer $\mathcal{B}=[(s_0,a_0,r_0,s_1), … ]$ can be used to collect all the (s,a,r,s') tuples generated at the end of each iteration. Then, in line 14 of Alg. 1, the gradients for updating the actor and critic networks can be obtained by first sampling a batch of (s,a,r,s') tuples from $\mathcal{B}$, followed by averaging over the gradients produced by each single tuple. Such batch-training technique is commonly used in RL training and more details can be found in [1].
>
> **Response to Question2**
>
> In the table below are preliminary runtime comparison results, between our method and state-of-the-art baselines, obtained using 50% missing MNIST digits generated following two different random masks. We could evaluate the runtime with more runs, and other datasets as well, in the camera ready version. All methods are evaluated on Nvidia RTX Quadro 6000 GPU and none exceeds the RAM limitation of the GPU (24GB). Note that the training epochs of GAIN and all the prediction models are calculated following [total_iterations / batch_size] since they were configured to set a max total iteration during training. It can be observed that our method requires a longer runtime since an actor and critic, both represented by neural networks, from the RL side need to be trained concurrently with the prediction model. However, the results summarized in Table 1,2 and 3 show that our method significantly outperforms baselines in both real-world and benchmark datasets -- indicating that it would be worth trading in runtime for significant performance improvement. Moreover, as one of the future directions, the efficiency could be further improved by designing a framework that can train multiple instances of RL agents in parallel, where the final policy can be obtained through voting/averaging -- for example, following ideas similar to [2].
>
> |        | Imputation train. total epoch | Imputation train. time per epoch | Imputation eval. time per epoch | Imputation total time | Prediction train. total epoch | Prediction train. time per epoch | Prediction total time | Overall total time  |
> |--------|-------------------------------|----------------------------------|---------------------------------|-----------------------|-------------------------------|----------------------------------|-----------------------|---------------------|
> | Our    | -                             | -                                | -                               | -                     | 23.27                         | 1937.41 ± 335.41                 | 45206.26 ±  2204.13   | 45206.26 ±  2204.13 |
> | MIWAE  | 2000                          | 70.86 ± 0.51                     | 1627.67 ± 6.90                  | 160114.64 ±  297.15   | 23.27                         | 0.70 ± 0.014                     | 16.33 ± 0.32          | 160130.97 ± 297.15  |
> | GAIN   | 58.18                         | 3.712 ± 0.012                    | 3.40 ±  0.097                   | 1474.27 ± 11.64       | 23.27                         | 0.70 ± 0.014                     | 16.33 ± 0.32          | 1490.6 ± 11.64      |
> | GP-VAE | 20                            | 278.03 ± 11.73                   | 117.23 ±  5.30                  | 5560.74 ±  33.37      | 23.27                         | 0.70 ± 0.014                     | 16.33 ± 0.32          | 5577.07 ± 33.37     |
>
> **References**
>
> [1] Lillicrap, Timothy P., Jonathan J. Hunt, Alexander Pritzel, Nicolas Heess, Tom Erez, Yuval Tassa, David Silver, and Daan Wierstra. "Continuous control with deep reinforcement learning." arXiv preprint arXiv:1509.02971 (2015).
>
> [2] Mnih, V., Badia, A.P., Mirza, M., Graves, A., Lillicrap, T., Harley, T., Silver, D. and Kavukcuoglu, K., 2016, June. Asynchronous methods for deep reinforcement learning. In International conference on machine learning (pp. 1928-1937). PMLR.

---

> ### Comment · Reviewer_azSY · 2021-11-18
> **Updated Review**
>
> Thanks for your well thought reply. I updated my recommendation to the accept side. I do not feel comfortable to go higher right now, though, as I am a little scared to overlook some major problem, due to my limited knowledge of the data missingness literature. Additionally, I still believe the cost and complicatedness of your algorithm is something that limits work to build upon it.
>
>
> Only vaguely related, a simple baseline proposal:
> Couldn't you simply use a Transformer on the inputs, as a Transformer, does not really care about missing entries or learns to handle them. See BERT for example. Of course, this does not scale very well to large image data, as each pixel would have to be it's own Transformer time step, but at least for MIMIC and OPHTHALMIC DATA, it should be possible, if I understand correctly, as the missingness patterns only involve a few feature groups.

---

> > ### Author Response · Authors · 2021-11-19
> > **Thank you for following up. Please see our responses below.**
> >
> > Thank you so much for providing ideas that could further enrich the experimental settings in our work! We could definitely include what you suggested as an additional baseline to the camera ready version. In the meantime, please note that the one of the current baselines, BRITS [1], follows a very similar idea — i.e., bidirectional RNNs are used to process input time-series, with attentions applied to the hidden states of RNNs. It optimizes over an imputation objective jointly with the prediction objective, and could be considered as the state-of-the-art method that can train prediction models end-to-end. However, we showed that our approach significantly outperforms BRITS on the MIMIC dataset.
> >
> > *----To further address your comments on the complicatedness and possibilities for future research to build on top----*
> >
> > Instead of proposing a single method that tries to solve the data missingness problem, we would envision that our work facilitates a *framework* illustrating how incomplete inputs can be tackled without introducing any imputation steps or objectives, by leveraging RL policies to guide the training of the prediction models. Specifically, we provide methodology on how to formulate RL environments (captured by MDPs) that can help achieve the goal. In the future, one could explore if there would be better ways to define the state and action space that could lead to faster convergence etc. More importantly, we have already showed that by incorporating (basic) representation learning terms into the reward function it can in general lead to improved performance. Therefore, reward shaping could be a topic that could be built directly on top of our framework, as the reward function we use is generic (i.e., negative of the prediction loss), where one could explore various ways of configuring the reward such that the resulting RL policy is expected to capture more information underlying missingness. On the other hand, reward shaping is also an important topic currently being explored in other domains that leverage RL techniques, such as robotics [2-4].
> >
> > If the paper gets accepted, we will devote our efforts to open source our code and datasets for future comparisons and reproducibility. As it could be observed from the code submitted to supplementary material that they are ready to be published. We, again, thank you for your efforts providing the invaluable feedback which really helps consolidate our work. Please do not hesitate to add any additional comments (if any) and we will try to address them in time!
> >
> > *References*
> >
> > [1] Cao, W., Wang, D., Li, J., Zhou, H., Li, L. and Li, Y., 2018. BRITS: Bidirectional Recurrent Imputation for Time Series. Advances in Neural Information Processing Systems, 31, pp.6775-6785.
> >
> > [2] Goyal, P., Niekum, S. and Mooney, R.J., 2019, August. Using Natural Language for Reward Shaping in Reinforcement Learning. In Proceedings of the 28th International Joint Conference on Artificial Intelligence.
> >
> > [3] Johannink, T., Bahl, S., Nair, A., Luo, J., Kumar, A., Loskyll, M., Ojea, J.A., Solowjow, E. and Levine, S., 2019, May. Residual reinforcement learning for robot control. In 2019 International Conference on Robotics and Automation (ICRA) (pp. 6023-6029). IEEE.
> >
> > [4] Botteghi, N., Sirmacek, B., Mustafa, K.A., Poel, M. and Stramigioli, S., 2020. On reward shaping for mobile robot navigation: A reinforcement learning and SLAM based approach. arXiv preprint arXiv:2002.04109.

---

### Official Review · Reviewer_he3p · 2021-11-02

**Correctness:** 4
**Technical Novelty And Significance:** 4
**Empirical Novelty And Significance:** Not applicable
**Recommendation:** 8
**Confidence:** 3

**Main Review:**

The general problem of doing predictions based on incomplete data is important. The ideas are simple and clearly presented. The experiments are comprehensive both in terms of comparing with new and classical methods and in terms of applying to real datasets to show applicability and toy datasets (MNIST) to understand qualitative performance. The limitations (e.g. that this is only applicable to MLPs and LSTMs) are also clearly stated.

**Summary Of The Paper:**

This paper proposes a method to do MLP / LSTM-based inferences/predictions based on incomplete data. While typical methods would impute and then predict, the proposed method predicts directly based on incomplete data. The method learns an “importance” matrix which is multiplied with the first weight matrix of the neural network. The learning is done using reinforcement learning, with the negative prediction error being the reward. This method is compared with new and classical prediction methods for incomplete data on a range of datasets and is shown to be the best most of the time.

**Summary Of The Review:**

N/A

---

> ### Author Response · Authors · 2021-11-16
> **Thank you for the feedback**
>
> Thank you for the feedback and we sincerely appreciate the time and effort spent on reviewing our manuscript. In case you have any further questions/comments, we would be very happy to address them for you!

---

### Decision · Program_Chairs · 2022-01-20

**Decision:**

Accept (Poster)

**Comment:**

The paper proposed an imputation free method to handle missing data by learning an input encoding matrix using RL with the prediction error as reward/penalty signal. Reviewers appreciate the interesting setup where RL is used to deal with missing data, and the method being imputation free. Three out of four reviewers (reviewer he3p, azSY, and 4Cb5) have raised concerns on the complexity of the proposed method, but it seems like all the reviewers see the strength of the work outweigh the weakness.